# Structural heterogeneity in the intrinsically disordered RNA polymerase II C-terminal domain

Bede Portz[1], Feiyue Lu[1,2], Eric B. Gibbs[3], Joshua E. Mayfield[4], M. Rachel Mehaffey[5], Yan Jessie Zhang[4,6], Jennifer S. Brodbelt[5], Scott A. Showalter[1,3] & David S. Gilmour[1]

RNA polymerase II contains a repetitive, intrinsically disordered, C-terminal domain (CTD) composed of heptads of the consensus sequence YSPTSPS. The CTD is heavily phosphorylated and serves as a scaffold, interacting with factors involved in transcription initiation, elongation and termination, RNA processing and chromatin modification. Despite being a nexus of eukaryotic gene regulation, the structure of the CTD and the structural implications of phosphorylation are poorly understood. Here we present a biophysical and biochemical interrogation of the structure of the full length CTD of *Drosophila melanogaster*, which we conclude is a compact random coil. Surprisingly, we find that the repetitive CTD is structurally heterogeneous. Phosphorylation causes increases in radius, protein accessibility and stiffness, without disrupting local structural heterogeneity. Additionally, we show the human CTD is also structurally heterogeneous and able to substitute for the *D. melanogaster* CTD in supporting fly development to adulthood. This finding implicates conserved structural organization, not a precise array of heptad motifs, as important to CTD function.

[1] Center for Eukaryotic Gene Regulation, Department of Biochemistry and Molecular Biology, The Pennsylvania State University, University Park, Pennsylvania, 16802, USA. [2] The Huck Institutes of Life Sciences. The Pennsylvania State University, University Park, Pennsylvania, 16802, USA. [3] Department of Chemistry, The Pennsylvania State University, University Park, Pennsylvania 16802, USA. [4] Department of Molecular Biosciences, University of Texas, Austin, Texas 78712, USA. [5] Department of Chemistry, University of Texas, Austin, Texas 78712, USA. [6] Institute for Cellular and Molecular Biology, University of Texas, Austin, Texas 78712, USA. Correspondence and requests for materials should be addressed to D.S.G. (email: dsg11@psu.edu).

RNA polymerase II (Pol II) transcribes primarily protein-coding genes in eukaryotes, a process regulated at initiation, elongation and termination, and concomitant with chromatin modification and RNA processing. The temporospatial regulation of transcription and co-transcriptional processes involves the concerted action of multiple factors bound to Pol II via a long, repetitive, intrinsically disordered C-terminal domain (CTD) comprised of heptads of the consensus sequence YSPTSPS[1]. In a process dubbed the CTD code, each non-proline residue of the repeat can undergo post-translational modification, and each proline can undergo *cis–trans* isomerization, creating a multitude of distinct binding sites for CTD-interacting proteins[2,3].

Despite its role as a regulatory nexus, little is known about the structure of the CTD, and most studies have focused on the structure of short CTD peptides. Such peptides have been shown to adopt β-turn structures when bound to CTD-interacting proteins, and β-spiral models of the CTD have been proposed that extrapolate β-turns across the entire length of the CTD[4]. However, nuclear magnetic resonance (NMR) and circular dichroism studies of CTD peptides suggest such turns are rare among the ensemble of CTD structures, and only populated to a high degree in turn-promoting solvents at low pH or in the context of circularized peptides[5–8]. Phosphorylation enables formation of additional hydrogen bonds that have been shown to contribute to turn structures in CTD peptides bound to factors, but phosphorylation fails to increase the β-turn propensity of unbound CTD peptides in solution[7,9]. Together, these data argue against models iterating turns across the full CTD in either the apo or phospho state. Such models are entropically unlikely, as they assume the CTD simultaneously adopts multiple low-probability turns. Additionally, β-spiral models fail to account for local structural variation potentially imparted by those heptads deviating from the consensus sequence, and such repeats comprise the majority of the CTD in developmentally complex organisms[10].

Existing structural information describing the full-length CTD is limited. The CTD, likely due to its flexibility, is absent from X-ray crystal structures of Pol II, but one study showed it may share space with the CTDs of adjacent Pol II molecules in the crystal, a space too small to accommodate extended structures, leading to a model that was sufficiently compact to fit into the space provided in the crystal, yet lacking the order associated with folded, globular structures[11]. The CTD is also absent from cryoEM derived models of elongating Pol II, again likely due to its flexibility[12]. Additional evidence for a compact CTD was obtained from cryoEM models of a GST-CTD fusion protein bound to the middle module of the multi-subunit transcriptional regulator known as the Mediator complex[13]. However, it is unknown to what extent binding to Mediator and possible CTD:CTD interactions resulting from GST-CTD dimers contribute to the observed compaction[13]. Earlier studies of negatively stained two-dimensional (2D) crystals of Pol II and Pol II mutants lacking the CTD revealed density differences attributed to a flexible CTD and linker occupying a space measuring ~100 Å in diameter[14]. The extent to which fixation on a solid surface alters the CTD ensemble in the 2D crystal studies is unknown, and like other X-ray crystallography studies, the proximity of CTDs from adjacent Pol II molecules in the 2D crystals could potentially alter the CTD ensemble through CTD:CTD interactions. One study by Corden and Zhang examined the structure of the mammalian CTD in solution and found the hydrodynamic radius of the CTD extends as a function of phosphorylation[15]. Consistent with extension was the observation by Laybourn and Dahmus that the CTD is more proteolytically labile in the phosphorylated state, suggesting conformations more accessible to CTD:protein interaction[16].

However, it is unclear from this work if the CTD was rendered more accessible by CTD phosphorylation alone, or in concert with possible interactions with proteins present in the HeLa extract component of the phosphorylated Pol II cleavage experiments, but absent from the unphosphorylated Pol II reactions[16]. Nevertheless, evidence supports an unphosphorylated CTD existing as a compact structural ensemble, capable of extending as a function of phosphorylation. Such an ensemble is consistent with emerging structural understanding of intrinsically disordered proteins (IDPs), which can be more compact than chemically denatured proteins of the same size, and can be structurally heterogeneous, possessing transient structural features[17–20].

The notion that IDPs can adopt semi-compact, heterogeneous ensembles is important in the context of CTD evolution. Much of our knowledge comes from studies on the CTD of *Saccharomyces cerevisae* that is unique in its sequence homogeneity, with nearly every heptad exactly matching the YSPTSPS consensus[10]. The CTD in mammals contains a consensus region proximal to the catalytic core of Pol II, and a non-consensus region at the distal end of the CTD composed of heptads differing primarily at the seventh position[1,10] (Fig. 1a,b). The CTD of *Drosophila melanogaster* is distinct from its human counterpart in that only two of forty-two heptads exactly match the consensus, though a similar consensus motif is identified[1,21] (Fig. 1a,b). Evolutionary differences in CTD composition raise the possibility that different motifs may serve to organize CTD:protein interactions via the co-evolution of factors that favourably recognize particular CTD motifs[10]. An alternative and not mutually exclusive role for non-consensus motifs could be the formation of structural heterogeneity, rendering particular regions of the CTD more accessible to protein interaction.

A model of CTD structure derived from data collected on the monomeric, full-length, metazoan CTD, under physiologically relevant buffer conditions, that relates global and local structure has been lacking. Here we present a biophysical and biochemical characterization of the CTD of *D. melanogaster,* which we identify as a structural homologue to its human counterpart. Using a combination of size exclusion chromatography (SEC) and Small Angle X-ray Scattering (SAXS), we show that the CTD undergoes moderate extension as a function of phosphorylation by the CTD kinase P-TEFb and a concomitant decrease in flexibility. In spite of the repetitive nature of the CTD sequence, we find structural heterogeneity in the CTD that is maintained in the phosphorylated state. Despite sequence differences between the CTD of *D. melanogaster* and *H. sapiens*, the human CTD is structurally similar to the fly CTD globally and both are structurally heterogeneous locally. Surprisingly, the human CTD is capable of functioning in place of the fly CTD *in vivo*, supporting the development of flies to adulthood. This finding implicates conserved structural organization, not a precise series of heptad motifs, as important to CTD function.

## Results

**A similar global CTD structure exists despite sequence differences**. CTD length and sequence complexity differ among organisms, with non-consensus heptads predominating in the longer CTDs of metazoans[1,10]. Nevertheless, aligning blocks of seven amino acids beginning with YSP for the CTDs of *D. melanogaster* and *H. sapiens* using Weblogo returns a similar consensus motif with most deviation occurring in the seventh position[21] (Fig. 1b). We sought to determine whether the presence of non-consensus repeats impacts the global structure of the CTD. Because the CTD naturally exists as a terminal extension emanating from the globular and acidic Pol II, and to

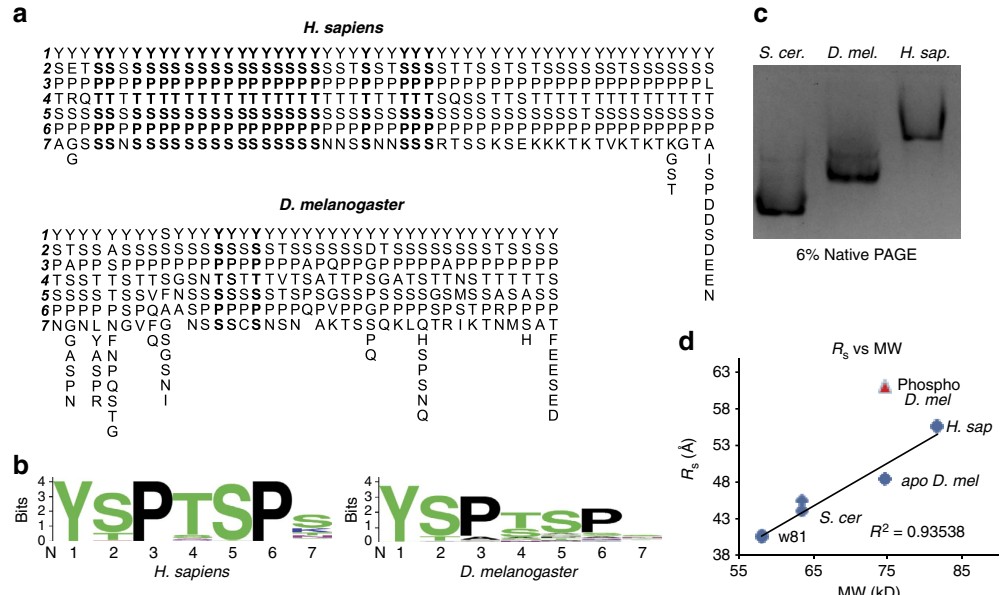

**Figure 1 | A similar global CTD structure exists despite sequence differences.** (**a**) The sequences of the CTDs of *D. melanogaster* and *H. sapiens* with each repeat beginning with YSP arranged vertically and the CTD oriented from left to right. Consensus heptads are in boldface. The *Drosophila* CTD has only two consensus heptads. (**b**) Motif conservation of fly and human CTDs. (**c**) Native gel electrophoresis of the *S. cerevisae*, *D. melanogaster* and *H. sapiens* MBP-CTD fusion proteins, with pIs of 5.57, 5.83, 5.93, respectively. Electrophoretic mobility scales with CTD length, suggesting structural homology. (**d**) $R_S$ versus molecular weight (MW) derived from size exclusion chromatography analysis of MBP-CTD fusions. Two replicates of each protein are plotted (several points appear as one because of overlapping values). $R_S$ is linearly related to MW, suggesting gross structural homology. Phosphorylation of the *D. melanogaster* CTD by P-TEFb increases $R_S$, causing the CTD to deviate from the line (red triangles).

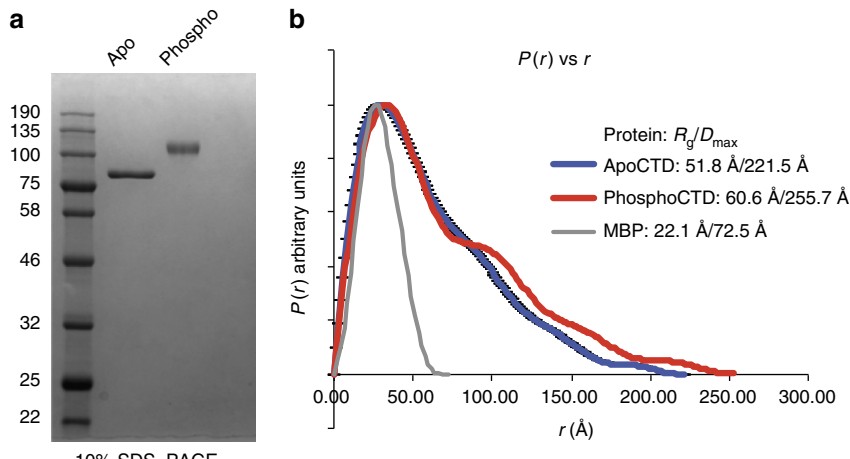

**Figure 2 | The CTD extends as a function of phosphorylation.** (**a**) Coomassie blue stained 12% SDS–PAGE gel of 600 ng of apo or phospho MBP-*D.mel*CTD used for SAXS experiments. (**b**) Pairwise distance distribution ($P(r)$ versus $r$) plots of the apo (blue line) and phospho (red line) MBP-*D.mel*CTD, and the CRYSOL generated scattering curve of MBP (grey line). The phospho CTD adopts a more extended conformation compared to the apo CTD, with increases in $R_g$ and $D_{max}$.

aid in purification and solubility, we expressed and purified the CTDs of *S. cerevisae*, *D. melanogaster*, and *Homo sapiens* as maltose binding protein (MBP) fusions and compared their mobility using native-PAGE. The relationship between CTD length and mobility for the three species suggests gross structural similarity despite differences in the relative proportion of non-consensus heptads (Fig. 1c).

To quantify the relationship between CTD length and overall structure more precisely, we compared the three CTDs using size exclusion chromatography (SEC), determining the Stokes radii ($R_S$) using a standard curve derived from natively folded proteins, and found a linear relationship between $R_S$ and molecular weight

(MW) for the CTD fusions[22,23] (Fig. 1d and Supplementary Fig. 1a,b). At some minimum CTD length, the MBP portion of the fusion could dominate the analysis, and it could be expected that this linear relationship would break down, confounding interpretation of the results. Thus, we included in this analysis a truncated version of the fly CTD, dubbed W81, shown to have functional defects *in vivo*[24]. Despite removing ~50% of the CTD, generating a CTD shorter than that of yeast, this fusion also remained in the linear range of the CTD curve (Fig. 1d). The relationship between $R_S$ and MW for all of the CTD fusions suggests that they are structural homologues despite sequence differences, sharing an extended conformation compared to

natively folded proteins of similar MW (Supplementary Fig. 1a,b). The correlation between $R_s$ and MW for the CTDs creates a CTD standard curve, with the *Drosophila* CTD at the center, that can be used to interrogate structural changes resulting from phosphorylation.

**Phosphorylation by P-TEFb extends the CTD.** Phosphorylation of the *D.mel*CTD by the CTD kinase P-TEFb lead to an apparent ~26% increase in the Stokes radius of the CTD, causing the phospho-CTD to fall off the CTD standard curve (Fig. 1d). To further interrogate differences in the structures of the apo and phospho CTD, we employed small angle X-ray scattering (SAXS), a high-precision technique well suited to the interrogation of IDPs in solution[25]. The CTD was subject to extensive phosphorylation and subsequent gel filtration chromatography, resulting in homogenous preparations of apo and phospho species with discrete mobility in SDS–PAGE and with no evidence of unphosphorylated protein remaining in the phosphoCTD sample (Fig. 2a). Using MALDI-TOF mass spectrometry, we determined that phosphorylation by P-TEFb under our conditions lead to an increase in MW of our MBP-*D.mel*CTD fusion of ~1.9 kDa, equivalent to the mass of ~25 phosphates (Supplementary Fig. 2a,b). Tandem mass spectrometry localizes this phosphate incorporation to primarily the fifth position, unless a heptad lacks a SP motif at positions 5–6 (Supplementary Fig. 2c), consistent with the *in vitro* preference for serine 5 for P-TEFb observed previously[26]. The density of phosphorylation; less than one phosphate per heptad with no heptads phosphorylated more than once, is in good agreement with *in vivo* observations, suggesting our biophysical analyses focus on a biologically relevant CTD phospho-isoform[27,28]. Guinier analysis of SAXS curves obtained across a range of concentrations for the apo and phospho CTD were in good agreement, showing no concentration effects across the dilution series in measured MW or radius of gyration ($R_g$), with phosphorylation leading to a ~15% increase in $R_g$ (Supplementary Fig. 3)[29,30] and an increase in MW of ~1.9 kDa, in excellent agreement with the MALDI-TOF data (Supplementary Fig. 2a,b).

Pairwise distance distribution functions for the averaged scattering curves from the apo and phospho MBP-*D.mel*CTD fusions were generated using AUTOGNOM (Fig. 2b)[31]. The $R_g$ values obtained via the $P_r$ versus $r$ analysis are slightly larger than obtained via Guinier analysis, which is not uncommon for IDPs, and the overall extension calculated in this analysis is in good agreement with that obtained from the Guinier fit (Fig. 2b)[32]. These analyses provide independent validation of the phosphorylation dependent extension we observe using SEC (Fig. 1d).

**The CTD is semi-compact.** The Flory chain model applied to disordered polypeptides relates protein length to $R_g$ via a power law, and has been used as a benchmark against which to compare the disorder, or unfoldedness, of a protein[33]. Applied only to the CTD portion of the fusion protein and omitting MBP, the Flory model for denatured proteins predicts an $R_g$ of ~60 Å (ref. 30). Notably, this is larger than the measured $R_g$ of 51.8 Å for the entire MBP-*D.mel*CTD fusion, which includes MBP, a globular protein with a mass exceeding that of the CTD and an estimated $R_g$ of ~22 Å. This comparison suggests that the CTD is considerably more compact than a denatured coil. Together, our measurements support a semi-compact CTD, more extended than a globular protein of the same molecular weight, but more compact than a denatured coil.

**Ensemble optimization models of the CTD.** To better conceptualize the CTD structure on the global scale, we generated ensemble models from the SAXS curves collected on apo and phospho MBP-*D.mel*CTDs using the Ensemble Optimization Method (EOM) 2.1, accounting for the contribution of the globular MBP[34,35]. This method generates 10,000 models for the CTD that are parsed using an algorithm to yield an ensemble whose average scattering curve agrees with the experimentally measured curve for our apo and phospho MBP-*D.mel*CTD fusions. These models yielded ensembles for the apo and phospho MBP-*D.mel*CTD with $R_g$ values of 50.91 and 60.98 Å, which are in good agreement with $R_g$ values derived from the $P(r)$ versus $r$ analysis (Figs 2b and 3). The resulting ensembles are depicted oriented by MBP and displayed with the 12-subunit RNA Pol II elongation complex model to allow for size comparison (Fig. 3)[35,36]. Notably, both compact and extended structures are found in the ensembles describing both the apo and phospho CTD, with a bias towards extended structures in the phospho state consistent with the extension observed in SEC and SAXS. SAXS data lack the resolution to interpret the models on the amino-acid or heptad scale, yet do provide a visual representation of the scale of the monomeric CTD in solution that is constrained by and in agreement with experimental data.

**The fly and human CTDs are structurally heterogeneous.** That the CTD is semi-compact and capable of undergoing extension as a function of phosphorylation prompted us to examine the structure at a more local level. Limited proteolysis experiments have been used to interrogate IDP structure, with regions of protease sensitivity and protection reflecting more unstructured and structured regions, respectively[37–39]. The repetitive nature of the CTD is ideally suited to this approach, as predicted cleavage sites are evenly distributed across the length of the CTD. If the CTD is either a completely disordered chain, or if it is structurally repetitive as β-spiral models predict, proteases would be predicted to cleave the CTD with equal probability along its length. Alternatively, local structural heterogeneity would manifest as regions of protease sensitivity or protection that can be mapped to regions of the CTD by comparison to a CTD ladder (Fig. 4a and Supplementary Fig. 4a–c). Chymotrypsin cleaves the peptide bond C-terminal to tyrosine and is predicted to cut the *Drosophila* CTD 43 times across its length[40]. Surprisingly, chymotrypsin yields a pattern of proteolysis characterized by a hypersensitive site towards the distal part of the CTD, and sensitivity located across the proximal region, with a protease

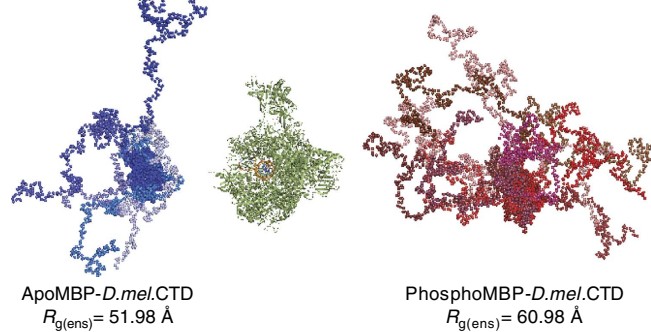

ApoMBP-*D.mel*.CTD
$R_{g(ens)}$ = 51.98 Å

PhosphoMBP-*D.mel*.CTD
$R_{g(ens)}$ = 60.98 Å

**Figure 3 | Ensemble optimization models depicting CTD extension.** Averaged apo and phospho MBP-*D.mel*CTD scattering data were modelled using EOM 2.1 from the ATSAS suite[31]. Individual models in the apo and phospho ensembles are shown in shades of blue and red, respectively, oriented by MBP and positioned adjacent to the 12 subunit RNA Pol II elongation complex (1Y1W.pdb) model (green) for scale.

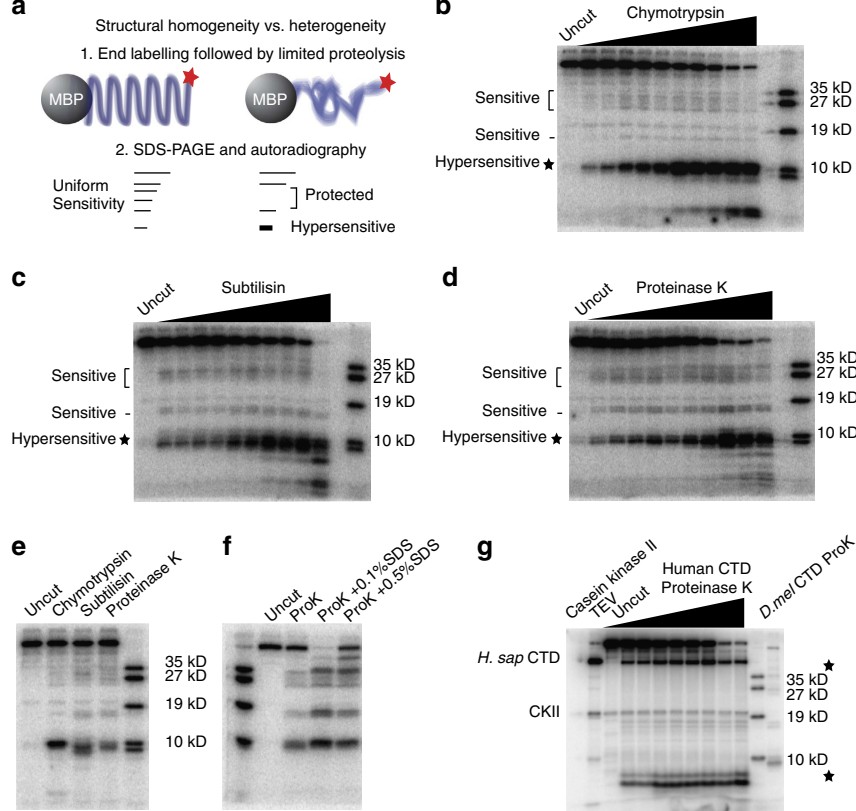

**Figure 4 | The CTD is structurally heterogeneous across its length.** (**a**) Experimental design: end-labelled MBP-*D.mel*CTD is subject to limited proteolysis, SDS-PAGE, and autoradiography. A completely unstructured, or alternatively, a structurally repetitive CTD is predicted to generate a uniform pattern of proteolytic fragments. A structurally heterogeneous CTD is predicted to give rise to a non-uniform pattern. (**b**) Limited proteolysis with chymotrypsin generates a non-uniform pattern of CTD fragments, with a hypersensitive site in the distal CTD, a sensitive region in the proximal CTD, and a largely protease insensitive region in the central CTD that is cleaved at only one site. The right most lane contains radiolabelled CTD fusion proteins with the molecular weights designated on the right of the panel. (**c,d**) Limited proteolysis with subtilisin or proteinase K reveal similar sites of sensitivity and protection. (**e**) A direct comparison of proteolytic fragments generated by three proteases shows a similar by not identical pattern of proteolysis. (**f**) Sodium dodecyl sulfate (SDS) alters the relative proteolytic sensitivity of the CTD to proteinase K, enhancing the sensitivity of the CTD at sites that generate bands near 35 kD and below 19 kD relative to the CTD ladder (0.1% SDS lane). 0.5% SDS renders the globular MBP portion of the fusion protein more susceptible to proteolysis, evidenced by the proteolytic fragment above the 35 kD CTD ladder band but below the intact fusion. (**g**) The human CTD is radiolabelled on the final acidic repeat by casein kinase II (CK2) (uncut lane). TEV cleavage to separate the human CTD from the MBP fusion demarcates the point below which protease sensitivity occurs in the CTD portion of the protein (TEV lane). Limited proteolysis with proteinase K reveals a distal hypersensitive site reminiscent of that observed in the *Drosophila* CTD (compare to *D.mel*CTD ProK lane), a protease hypersensitive proximal site near MBP and a central region that is largely protease insensitive. Both the human and fly CTDs share distal protease hypersensitivity, proximal sensitivity that is localized to a discrete region in the human CTD, and a more central region of the CTD that is less protease sensitive.

insensitive region spanning the central part of the CTD that is cut in only one location (Fig. 4b). One interpretation of this result is that the CTD is not uniformly accessible to protein-sized factors due to local structural variation. Alternatively, this result could be due to sequence variation in the *Drosophila* CTD giving rise to non-uniform recognition of potential cleavage sites. To distinguish between these hypotheses, we subjected the MBP-*D.mel*.CTD to limited proteolysis with subtilisin, a protease possessing similar specificity to chymotrypsin but that is structurally unrelated, and proteinase K, predicted to cut the CTD at 100 sites[40]. Strikingly, the proteolytic pattern generated by the additional enzymes recapitulates that of chymotrypsin (Fig. 4c,d). To determine if the MBP may be altering the structure of the CTD, we repeated the limited proteolysis of the CTD fused to glutathione S-transferase (GST), observing the same protease sensitivity when compared to our MBP-CTD experiment, arguing against structural influence on the CTD by the fusion protein (Supplementary Fig. 4d) Comparing directly the digested products from all three proteases at like titration points reveals

a similar but not identical pattern of proteolysis (Fig. 4e). We also performed limited proteolysis in the presence of the denaturant sodium dodecyl sulfate in concentrations tolerated by proteinase K[41]. SDS alters the pattern of proteolysis, altering the distribution of sensitivity relative to the no-SDS control, consistent with disruption of local structure in the CTD substrate[41] (Fig. 4f). Thus, we favour the hypothesis that the CTD is structurally heterogeneous and not uniformly accessible to protein interaction across its length and that this organization is an intrinsic feature of the CTD.

Despite a large disparity in the proportion of the fly and human CTD comprised of consensus heptads, 5% versus 41%, respectively, our native gel and size exclusion chromatography results indicated gross structural similarity. This led us to consider if the human CTD may also share the local structural heterogeneity we identify in the fly CTD. We subjected the human CTD to limited proteolysis with proteinase K, predicted to cleave the human CTD at 122 locations. Surprisingly, we observed a similar pattern of protease hypersensitivity between fly

and human, with the human CTD also exhibiting a hypersensitive site in the distal region, in addition to a hypersensitive site in the proximal region near where the collection of sensitive sites resides in the *Drosophila* CTD, suggesting that conservation in CTD structure between fly and human exists despite length and sequence differences (Fig. 4g).

**Structural heterogeneity persists after phosphorylation.** In light of the findings that the CTD undergoes extension as a function of phosphorylation and is structurally heterogeneous in the apo state, we sought to determine if the CTD is locally reorganized in the phospho state. To address this question, we performed limited proteolysis on the phospho CTD using proteinase K (Fig. 5a). The phosphorylated CTD retains a non-uniform pattern of cleavage with the resulting proteolytic fragments having shifted mobility due to phosphorylation, indicating local structural heterogeneity is maintained (Fig. 5a). However, the sensitivity of the various regions changes relative to each other. For example, at digestion points with minimal amounts of protease, products in the lower region of the gel are more abundant than products near the central region for the apo CTD (Fig. 4d) whereas they are nearly equivalent in the case of the phosphorylated CTD (Fig. 5a). Overall, phosphorylation of the CTD results in more even level of cutting at all of the labile sites.

**CTD phosphorylation increases protein accessibility.** The more even cleavage of protease accessible sites across the phospho CTD prompted us to test whether phosphorylation increases CTD accessibility. We subjected a mixture of apo and phospho CTD to limited proteolysis and compared the level to which each isoform was digested as a function of increasing protease concentration (Fig. 5b). We quantified three experimental replicates, comparing the amount of intact CTD remaining at each protease concentration. This revealed a moderate increase in the level of proteolysis of the phospho CTD, which is digested past the point of single hit kinetics ($>50\%$ cleavage[42]) at a lower protease concentration, consistent with a moderate increase in accessibility as a function of phosphorylation (Fig. 5c).

**The CTD stiffens as a function of phosphorylation.** SAXS data can be used to make qualitative comparisons of the flexibility of proteins and to monitor flexibility changes after post-translational modification[43]. Kratky plots enable comparison of protein flexibility, and indicate that the CTD becomes less flexible when phosphorylated[43] (Fig. 6a and Supplementary Fig. 5). To further validate this result we generated Porod–Debye and Kratky–Debye plots (Fig. 6b,c) from the apo and phospho CTD scattering curves. These analyses truncate the high $q$ region of the data and thus are less prone to potential artifacts of buffer subtraction[43]. The asymptotic rise in the Porod-Debye plot exhibited by the apo CTD curve is characteristic of flexible proteins and is diminished for the phospho CTD (Fig. 6b)[43]. Further, the plateau observed for the apo CTD in the Kratky–Debye plot is lost for the phospho CTD (Fig. 6c). The divergent behaviour of the apo and phospho CTD between the Kratky–Debye and Porod–Debye plots and the differences within each plot for the apo and phospho CTD are indicative of a phosphorylation-dependent decrease in flexibility[43].

**The human CTD can function in place of the fly CTD *in vivo*.** The structural similarity but sequence differences between the fly and human CTDs, coupled with the genetic tractability and developmental complexity of *Drosophila*, emboldened us to test the hypothesis that the evolution of non-consensus CTD repeats imparted important lineage specific functionality[1,10]. The high degree of conservation of the CTD sequence among the twelve species of *Drosophila* (Fig. 7a) and separately among mammals contrasts with the sequence divergence between the two phyla (Fig. 1a)[44]. This suggests that within various lineages, specific CTD sequences arose and were maintained under selective pressure, perhaps co-evolving with CTD interacting factors in a way that could recruit new activities to RNA Pol II or organize multiple CTD:factor interactions across the CTD. To test the importance of a precise CTD amino acid sequence in a developmental context, we used RNAi to knock down endogenous Rpb1 in the developing fly wing and determined whether coincident ectopic expression of an RNAi resistant Rpb1 with either the normal *Drosophila* CTD or the human CTD

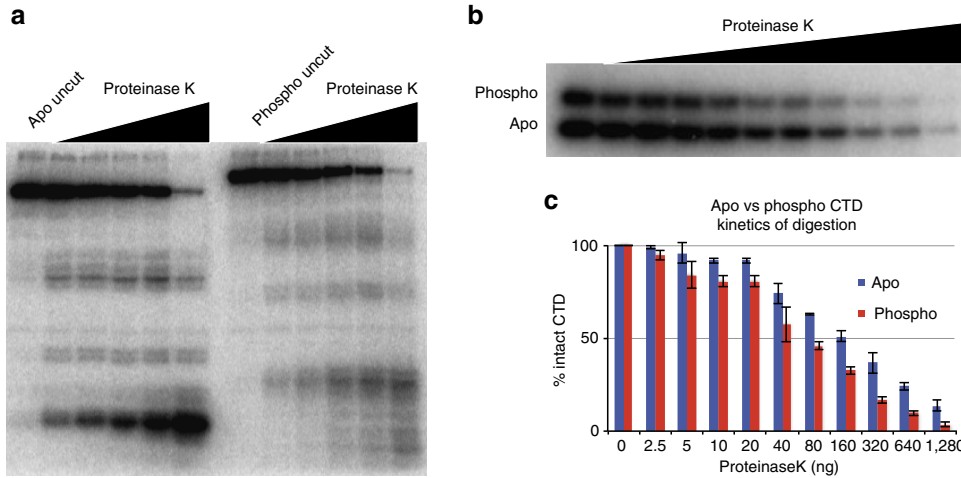

**Figure 5 | Local structural heterogeneity is maintained in the phospho CTD but the phospho CTD is more accessible to protein interaction. (a)** Limited proteolysis of the phospho CTD shows an altered pattern of proteolysis. The hypersensitivity of the distal site is reduced, and the pattern of proteolysis is more evenly distributed among the sensitive sites. The proximal sensitive region, and central protected region are preserved after phosphorylation, with proteolytic fragments shifted relative to the apo fragments due to phosphorylation across the CTD. **(b)** Representative limited proteolysis of a mixture of apo and phospho CTD to compare protease accessibility. **(c)** Average of three replicates of the apo (blue bars) and phospho (red bars) mixing experiments quantified as the percentage of intact CTD remaining at each protease concentration. 100–50% intact CTD is the single hit kinetics range of the experiment[42]. Error bars depict s.e.m.

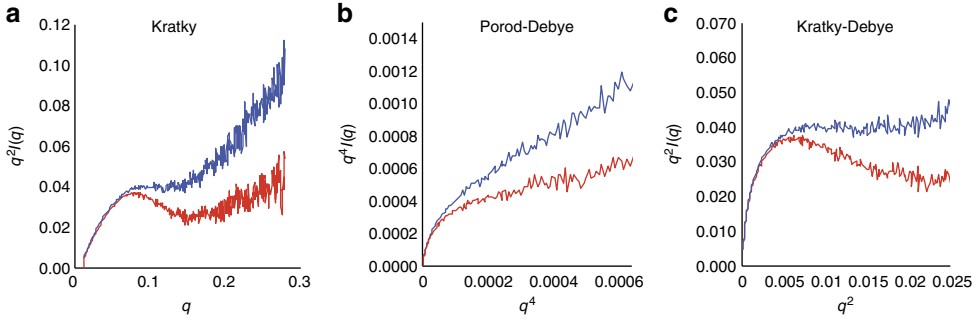

**Figure 6 | Phosphorylation stiffens the CTD. (a)** Kratky Plot ($q^2 \times I(q)$ versus $q$) of averaged apo (blue) and phospho (red) MBP-CTD scattering curves. The more gradual rise in the high q region of the plot for the phospho-CTD than the apo-CTD indicates that the phospho-CTD is less flexible than the apo CTD. **(b,c)** Porod–Debye ($q^4 \times I(q)$ versus $q^4$) and Kratky Debye ($q^2 \times I(q)$ versus $q^2$) plots of the apo (blue) and phospho (red) scattering curves. The increased rise in the Porod-Debye plot for the apo-CTD compared to phospho-CTD indicates a phosphorylation dependent decrease in flexibility, as does the loss of the plateau in the Kratky–Debye plot for the phospho-CTD.

would revert phenotypes caused by depleting the endogenous Rpb1. RNAi-mediated depletion of Rpb1 in the wing caused a wing undergrowth phenotype that could be reverted to wild-type by co-expressing RNAi resistant *Drosophila* Rpb1 (Supplementary Fig. 6a,b). Coincident expression of a defective version of *Drosophila* Rpb1 called W81, which has approximately half of the CTD deleted from the region distal to the body of Pol II, only partially restored wing growth, showing that the assay is sensitive enough to reveal a gradient of phenotypes (Supplementary Fig. 6c). Remarkably, expression of RNAi resistant Rpb1 harbouring the human CTD fully reverts the undergrowth phenotype caused by the Rpb1 RNAi (Supplementary Fig. 6d).

Motivated by the results in the wing, we sought to determine whether RNAi-resistant versions of wild-type or humanized Rpb1 could rescue the lethality caused by ubiquitously expressing Rpb1 RNAi in the entire animal and throughout development. In these experiments, flies ubiquitously expressing the transcriptional activator GAL4 under the control of the actin gene promoter are mated to flies harbouring both RNAi against Rpb1 and an additional Rpb1 derivative rendered resistant to RNAi via synonymous mutations, both of which are expressed in the presence of GAL4 (Fig. 7b). Progeny not expressing GAL4, and thus not expressing Rpb1 RNAi, have curly wings (Fig. 7b). Progeny expressing GAL4, and thus expressing Rpb1 RNAi and an RNAi resistant derivative, have straight wings. As expected, approximately 50% of the progeny had straight wings when the Actin-GAL4 driver was crossed to *yw* control flies lacking Rpb1 RNAi (Fig. 7c, *yw*). In contrast, none of the progeny had straight wings when the Actin-GAL4 driver was combined with Rpb1 RNAi alone, indicating lethal levels of Rpb1 knockdown are achieved using this system (Fig. 7c, Rpb1i). In addition, no straight-winged progeny were present when an RNAi-sensitive version of *Drosophila* Rpb1 was over-expressed along with the Rpb1 RNAi (Fig. 7c, Rpb1$^{wt}$ [sens],Rpb1i) indicating that the level of RNAi knockdown is robust enough to kill the flies even when Rpb1 is ectopically overexpressed in addition to endogenous levels. In stark contrast, approximately 50% of the progeny have straight wings when an RNAi-resistant form of *Drosophila* Rpb1 is present, indicating rescue of Rpb1 RNAi by the ectopically expressed derivative (Fig. 7c, Rpb1$^{wt}$ [res], Rpb1i). Remarkably, the RNAi-resistant Rpb1 harbouring the human CTD also rescues the lethality caused by the Rpb1 RNAi (Fig. 7c, Rpb1$^{hu}$, Rpb1i). Molecular analysis of the straight winged progeny shows that the RNAi reduces the endogenous Rpb1 mRNA by about 20 fold relative to control flies (Supplementary Fig. 7a). Moreover, comparable levels of each ectopically expressed FLAG-tagged Rpb1 are being expressed (Supplementary Fig. 7b,c).

We next used immunostaining of polytene chromosomes from third instar larvae to further validate that the humanized Rpb1 is functioning normally. Because only ~50% of offspring from our crosses are expressing FLAG-tagged Rpb1 derivatives and RNAi, immunostaining chromosomes obtained from multiple larvae generates instances where chromosomes staining positively for FLAG-Rpb1 were located next to chromosomes from individuals not expressing the FLAG-Rpb1 and Rpb1 RNAi. The latter fail to stain with FLAG antibody but are visible using DNA stain (Fig. 7d, left and middle panels). The positive FLAG staining shows that the ectopically expressed Rpb1 derivatives harbouring either the wild-type or the human CTD associate with chromosomes (Fig. 7d, middle panels). Higher magnification images of chromosomes staining positively for FLAG-Rpb1 show that the ectopically expressed wild-type and humanized Rpb1 co-localize on chromosomes with the Rpb3 subunit of Pol II (Fig. 7d, right panels). Together, these data indicate that ectopically expressed Rpb1 with the human CTD is expressed, co-localizes on chromosomes with another subunit of Pol II, and is able to support growth and development when endogenous Rpb1 is knocked down by ~95% to lethal levels.

We next sought an independent test of the ability of the human CTD to function in place of the fly CTD that does not rely on RNAi knockdown. To this end, we employed the Rpb1$^{G0040}$ fly line, which contains a P-element insertion in the *Rpb1* gene that causes early embryonic lethality. This mutation is maintained over a balancer chromosome, FM7c (Fig. 7e)[45]. We recombined our *Rpb1* transgenes with a *daG32-Gal4* transgene so that there would be ubiquitous expression of the *Rpb1* transgene. In triplicate, eight virgin females from the lethal Rpb1 mutant line were crossed to eight males ectopically expressing Rpb1, or to *yw* as a negative control (Fig. 7e and Supplementary Table 1). Each cross yields three possible types of males, which can be distinguished phenotypically (annotated A, B and C in Fig. 7e). 'A' males carry a wild-type endogenous Rpb1 allele (Rpb1$^+$) on FM7c and are identified by the narrow eye phenotype ($B^I$). 'B' males result from ectopic expression of humanized (Rpb1$^{hu}$) or wild-type (Rpb1$^{wt}$) Rpb1 complementing the early embryonic lethality of the Rpb1$^{G0040}$ allele, and are identified by normal pigmented bodies ($y^+$). The presence of B males from both the human and wild-type CTD crosses establishes that these derivatives of Rpb1 can complement the early embryonic lethal allele (Fig. 7e, B offspring). 'C' males are the result of non-disjunction and are characterized by yellow bodies ($y^-$). The crosses to *yw* failed to complement (Fig. 7e, B offspring) and

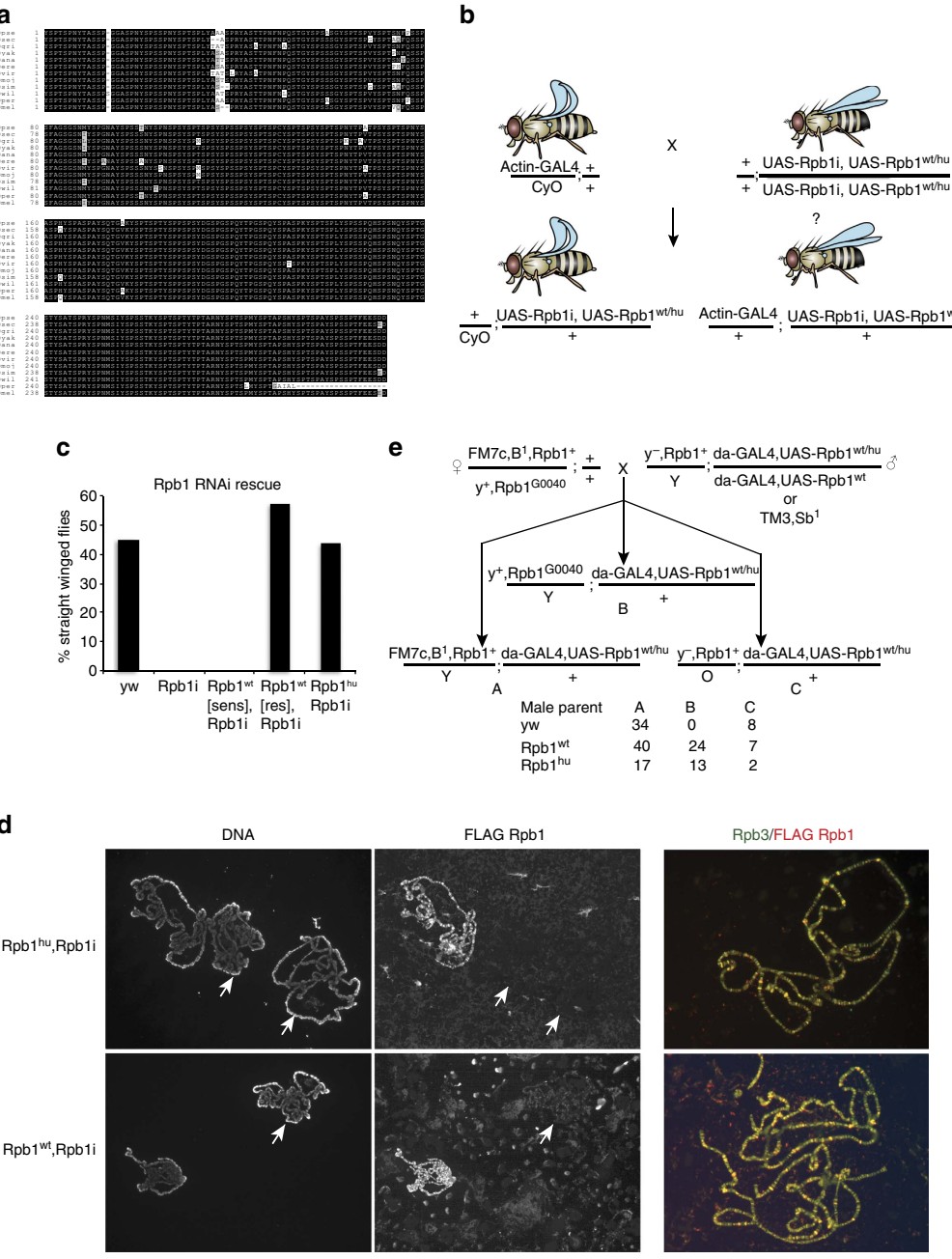

**Figure 7 | The human CTD can function in place of the *Drosophila* CTD *in vivo*.** (**a**) CTD amino acid sequence alignment from 12 species of *Drosophila*. (**b**) Experimental design of the human CTD rescue experiment. *UAS-Rpb1i* is a GAL4 inducible transgene that produces an RNAi against endogenous Rpb1. *UAS-Rpb1* is a GAL4 inducible transgene that encodes a FLAG-tagged, RNAi-resistant derivative of Rpb1. Fly images were created with Genotype Builder[54], (**c**) Results of crosses to test if ectopically expressed derivatives of Rpb1 rescue the lethality caused by ubiquitously expressing Rpb1 RNAi. Each bar on the histogram indicates the percentage of progeny with straight wings. *yw*: *ActGAL4/CyO* × *yw* ($n=137$ progeny); Rpb1i: *ActGAL4/CyO* × UAS-*Rpb1i* ($n=86$ progeny); Rpb1$^{wt}$[sens],Rpb1i: *ActGAL4/CyO* × UAS-*Rpb1i,UAS-Rpb1$_{sen}$* ($n=102$ progeny); Rpb1$^{wt}$[res],Rpb1i: *ActGAL4/CyO* × UAS-*Rpb1i,UAS-Rpb1$_{res}$* ($n=91$ progeny); Rpb1$^{hu}$,Rpb1i: *ActGAL4/CyO* × UAS-*Rpb1i, UAS-Rpb1$^{hu}$* ($n=78$ progeny). (**d**) Immunofluorescence of polytene chromosomes from salivary glands of third instar larvae. Larvae were derived from mating *ActGAL4/CyO* and *UAS-Rpb1i, UAS-Rpb1* parents, so half of the larvae ectopically expressed *Drosophila* or humanized Rpb1 (Left and Middle panels). FLAG Rpb1 detected on chromosomes co-localizes with the Rpb3 subunit of Pol II (Right panel). Arrows highlight chromosomes from individuals not expressing FLAG Rpb1 variants (*CyO/+; UAS-Rpb1i,UAS-Rpb1/+*). (**e**) Genetic complementation assay mating scheme and results. Complementation was scored in males, which have a single copy of the X chromosome on which the endogenous Rpb1 gene resides. Female flies carrying the early embryonic lethal Rpb1 mutant allele, G0040 are crossed to male flies expressing Rpb1$^{wt}$ or Rpb1$^{hu}$ under the control of da-GAL4, or to *yw* control flies. Crosses were carried out in triplicate with 8 males and 8 female parents per cross and flies were grown at 24C and 70% humidity. Shown in **e** are the total counts of male progeny; counts from individual vials are shown in Supplementary Table 1. 'B' males indicate that the ectopically expressed Rpb1 complements of the lethal allele. We observe no 'B' flies from the *yw* control crosses.

never yielded 'B' offspring, as it lacked an ectopically expressed Rpb1 transgene to complement the embryonic lethal Rpb1[G0040] allele, showing that the G0040 allele confers lethality in this assay.

These results show that the human CTD can function in place of the highly conserved *Drosophila* CTD when the endogenous polymerase is present at levels too low to support viability. This finding has two important implications. First, the results argue against the requirement for a precise CTD sequence even in a complex developmental context. Second, the results argue that the conserved structural organization of the fly and human CTD is important for CTD function.

## Discussion

Existing models of CTD structure were generated by iterating structural motifs across the length of the CTD. These models leveraged the best available structural data from CTD peptides, but evidence for turn motifs in unbound CTD peptides in physiologically relevant buffer conditions was lacking[5–9]. Evidence for a compact but flexible CTD was more compelling, but was potentially biased by CTD:CTD or CTD:factor interactions or fixation conditions required by the various experiments[11,13,14]. Here we present biophysical data on the monomeric full-length CTD obtained with both size exclusion chromatography and small angle X-ray scattering that supports a model for CTD structure that we describe as a compact random coil: more extended than predicted for a natively folded protein of the same mass, but significantly more compact than a denatured coil.

The primary role of the CTD is thought to be that of a scaffold protein capable of recruiting proteins to Pol II involved in transcription initiation, elongation, termination, and co-transcriptional processes such as mRNA capping, splicing and polyadenylation; further, these interactions occur in a temporospatially regulated manner[2,3]. Compactness may protect the CTD from imprecise post-translational modification and factor recruitment out of phase with the transcription cycle. Moreover, compactness could also facilitate the transcription process by preventing the CTD from becoming entangled with the DNA, RNA or other proteins. Using a series of proteases as protein scale probes, we interrogated the structure of the CTD and found a pattern of sensitivity and protection we interpret as evidence of a heterogeneous CTD structure containing regions with varying degrees of disorder. We favour this interpretation over the alternative hypothesis that variation in the primary amino acid sequence of the CTD gives rise to discontinuous proteolysis. First, three separate proteases with variations in structure and specificity all recapitulate a similar but not identical pattern of proteolysis, and these shared patterns result from proteolysis at only a subset of the total cleavage sites predicted by the specificity of each enzyme (Fig. 4b–e). This suggests that each protease first interacts with the accessible regions of the CTD, before cleaving at subtly different sites therein as a function of their individual specificities. Second, the denaturant SDS, known to stimulate proteolysis through the disruption of protein structure, alters the relative pattern of proteolysis of the CTD by proteinase K (Fig. 4f). That some elements of the proteolytic sensitivity are altered by a denaturant suggests we are detecting a bona fide structural phenomenon. Finally, despite differences in primary amino acid sequence, the human CTD, like the fly CTD, also displays a discontinuous pattern of proteolysis with a hypersensitive site in the distal region of the CTD and a proximal protease sensitive region, suggesting structural heterogeneity is a feature common to both organisms (Fig. 4g).

The finding that fly and human CTD exhibited similar patterns of proteolysis lead us to test a long standing hypothesis: that the co-evolution of non-consensus heptads and CTD binding proteins imparts lineage specific functionality, via added specificity of CTD binding interactions originating from the co-evolution of the CTD with its binding partners[1,10]. Such specificity could in theory serve to organize multiple proteins on a single CTD or modulate binding specificities among competing factors. Our ability to replace the *D. melanogaster* CTD sequence, which is highly conserved among *Drosophila* species, with the human CTD sequence bearing different heptads but a similar structure, suggests the CTD sequence is not organizing binding interactions in an essential way, even in a developmentally complex context (Fig. 7c–e). Instead, we posit that structural organization may serve to preferentially expose particular regions of the CTD that in turn could control the placement of post-translational modifications and organize binding interactions.

Phosphorylation leads to extension of the CTD that is modest as a percentage of its initial size (Figs 1d and 2), which results in modestly increased protein accessibility as measured by a comparison of proteolytic sensitivity between the apo and phospho protein (Figs 4 and 5). Analysis of the SAXS data indicates that the phosphorylated CTD is less flexible than the unphosphorylated CTD (Fig. 6) Meanwhile, local structural heterogeneity is maintained (Fig. 5a). These observations are interesting in the context of those CID:CTD co-crystal structures of factors that favour binding phosphorylated CTD peptides without directly recognizing the phosphate via hydrogen bonding[4]. Together these observations support a model for CTD binding in which multiple effects of phosphorylation may in concert facilitate factor binding. First, the CTD transitions from a polyampholyte to a polyanion, altering its affinity for factors, potentially as a function of phosphate density[46]. Extension de-protects the CTD, making it more accessible to binding. A less flexible CTD with reduced conformational entropy may provide a more binding-competent dock by depleting the CTD ensemble of conformers incompatible with binding, thereby reducing the entropic penalty for binding[47]. A compelling paradigm for phosphoCTD recognition may be the phospho-protein Sic1, which like the CTD is compact, intrinsically disordered, comprised of a repetitive and low complexity sequence, and is subject to phosphorylation at multiple repeats giving rise to multivalent binding[46,47]. Cdc4 can recognize phospho Sic1 on any one of multiple phosphorylated repeats, but only after a certain threshold of phosphorylation is reached that achieves adequate charge density[46] and possibly reduced conformational entropy[47] which results in binding 'ultrasensitivity', binding characterized by extremely high Hill coefficients. Likewise, analogous changes in the CTD could tip the balance in favour of binding and explain how a CTD dynamically sampling extended structures avoids unregulated modification and premature binding in the presence of hundreds of interacting proteins in the nuclear compartment.

## Methods

**Sequence motif generation.** The amino acid sequences of the Pol II CTDs of *D. melanogaster* and *H. sapiens* were entered into WebLogo as chains of seven amino acids, beginning with YSP[21]. For compatibility with the tool, repeats longer than seven amino acids were trimmed, those shorter were extended with 'X.'

**Cloning MBP-CTD plasmids.** PCR amplicons encoding the *S. cerevisae*, *D. melanogaster*, and *H. sapiens* CTD with either 6HIS or FLAG-STREP (DYKDDDDK-WSHPQFEK) tags were inserted into the XbaI digested pMalX vector using infusion cloning (Clontech) and transformed into Stellar competent *E. coli* (Clontech). CTD sequences were verified by sequencing.

**Expression of MBP-CTD proteins.** pMalX vectors containing the CTD fragments with either C-terminal 6HIS tags or C-terminal FLAG-STREP (DYKDDDDK-WSHPQFEK) tags were transformed into BL21DE3 *E. coli* cells and grown in LB

media with 0.2% dextrose and 100 µg ml$^{-1}$ ampicillin at 37 °C shaking at 250 r.p.m. to an optical density at 600 nm of ~0.5. Protein expression was induced with 300 µM IPTG, and proteins were expressed overnight at 11 °C with shaking at 250 r.p.m. Cells were collected by centrifugation for 10 min at 3,000g, flash frozen and stored at −80 °C. Thawed pellets were lysed by sonication in 20 ml per 1 l culture volume of Binding Buffer (50 mM HEPES pH 7.5, 500 mM NaCl, 10% glycerol, 1% NP-40, 2.5 mM imidazole, 2.5 mM β-mercaptoethanol) plus protease inhibitors.

**Purification of MBP-CTD proteins.** Cell lysates were centrifuged at 100,000 × g for 30 min and the clarified lysates were bound to TALON superflow resin (Clontech) for 6HIS tagged proteins or Streptactin superflow resin for FLAG-STREP (iba) in batch for 30 min at 4 °C. Resin was centrifuged at 500 g for 5 min and washed with 20–40 column volumes (CV) of Wash Buffer A (50 mM HEPES pH 7.5, 500 mM NaCl, 10% glycerol, 1% NP-40, 10 mM imidazole, 2.5 mM β-mercaptoethanol) in batch. Resin was centrifuged as above, resuspended in 6 CV of wash buffer A and poured into a column and washed with an additional 3 CV of wash buffer. Protein was eluted in 8 CV of Elution Buffer (50 mM HEPES pH 7.5, 500 mM NaCl, 10% glycerol, 1% NP-40, 200 mM imidazole, 2.5 mM β-mercaptoethanol; for FLAG-STREP tagged proteins, this buffer also contained 2.5 mM desthiobiotin). Eluted protein was passed twice over an amylose resin column (0.5 ml column resin per l cell culture). The resin is rinsed twice with 10CV of wash buffer B (50 mM HEPES pH 7.5, 150 mM NaCl, 10% glycerol, 5 mM DTT, 1 mM EDTA) and eluted in CTD buffer (50 mM HEPES pH 7.5, 150 mM NaCl, 10% glycerol, 5 mM DTT, 10 mM maltose, 1 mm EDTA)

**Expression and purification of D. melanogaster P-TEFb.** 600 ml of Sf9 cells were grown in suspension in baffled flasks at 27 °C in Sf-900 II SFM media (Thermo Fisher Scientific), diluted to 1.5 million cells per ml, and infected with 1/10 culture volume of P-TEFb virus (a gift from J.T. Lis) and incubated at 27 °C with shaking at 75 r.p.m. 72 hrs after infection, the cells were collected at 500 g for 5 min and resuspended in P-TEFb binding buffer (50 mM HEPES pH 7.5, 500 mM NaCl, 10% glycerol, 1% IGEPAL, 2.5 mM Imidazole and 2.5 mM β-mercaptoethanol and protease inhibitors) and lysed in a dounce homogenizer. Lysate was centrifuged at 100,000 g for 30 min at 4 °C. Clarified supernatant was bound in batch to 0.5 ml TALON resin for 30 min at 4 °C, resin was pelleted at 250 g for 5 min, washed with wash buffer, pelleted as before, washed and decanted into a gravity column and protein was eluted in 500 mM NaCl, 50 mM HEPES pH 7.5, 10% glycerol, 1% NP-40, 2.5 mM β-mercaptoethanol and 200 mM imidazole.

**Phosphorylation and purification of MBP-CTD.** MBP-D.melCTD6HIS was purified as described above, concentrated with a Vivaspin 30,000 MWCO concentrator (GE Life Sciences) to ~10 mg ml$^{-1}$ and kinased in CTD buffer supplemented with 10 mM MgCl$_2$, 10 mM ATP and P-TEFb at 24 °C for ~48 h with additions of 1/10th reaction volume 100 mM ATP and 10 mM MgCl$_2$ at ~8–12 h intervals for the duration of the reaction to restore ATP concentrations to ≥10 mM, while maintaining MgCl$_2$ concentration at 10 mM. The kinase reactions were loaded on a Superose6 10/300 column (GE Life Sciences) in CTD buffer and the monomeric peak fractions were collected and concentrated to ~1–2 mg ml$^{-1}$ using a Vivaspin 30,000 MWCO concentrator previously rinsed with CTD buffer. For SAXS, exact concentrations were obtained via UV absorbance at 280 nm and the molar extinction coefficient of the fusion protein (124,460 M$^{-1}$ cm$^{-1}$). Dilutions for SAXS were done in the same preparation of CTD buffer used for the size exclusion chromatography step of the sample preparation.

**MALDI-TOF intact mass analysis.** Proteins at an initial concentration of 1 mg ml$^{-1}$ were diluted four-fold using an aqueous solution of trifluoroacetic acid (TFA). The final concentration of TFA was maintained at 0.1% and the pH was verified to be <4 using pHydrion pH test paper (Sigma). Samples were desalted and concentrated using ZipTips with 0.6 µl C$_4$ resin (Millipore) according to manufacturer instructions and eluted in acetonitrile/water/TFA (60/40/0.1 v-v:v). 0.5 µl of desalted sample was spotted and overlaid with 0.5 µl sinapinic acid matrix solution (saturated in acetonitrile/water 50/50 v-v) onto a metal sample plate. Spots were allowed to air dry at room temperature until crystals formed. MALDI-TOF spectra were acquired on a Voyager-DE PRO (Applied Biosystems) instrument using the preloaded myoglobin_linear parameters in negative mode. Laser intensity and number of flashes were manually adjusted to provide the greatest signal over noise. Spectra for the apoMBP-D.melCTD6HIS and phosphoMBP-D.melCTD6HIS were acquired independently from neighboring sample spots to reduce spectra variation. Spectra were analysed and post-processed in Data Explorer (Applied Biosystems). Spectra were baseline corrected and the Noise Removal utility was used to remove noise within two standard deviations. Masses were determined via single point calibration using the theoretical mass of unmodified apoMBP-D.melCTD6HIS (74,704.13 Da) as the reference standard for the highest intensity peak of the post-processed apoMBP-D.melCTD6HIS spectra. The resultant calibration was applied to the post-processed phosphoMBP-D.melCTD6HIS spectra to determine molecular weight.

**Mass spectrometry for phospho-site identification.** Reduction of MBP-D.melCTD samples using 5 mM dithiothreitol (60 min at 55 °C) was followed by alkylation of reduced cysteines with 15 mM iodoacetamide (30 min at room temperature in the dark). Samples were then diluted into 100 mM Tris–Cl (pH 8) containing 10 mM CaCl$_2$ and digested overnight at room temperature with chymotrypsin (1:50 enzyme to substrate ratio). Digests were desalted on C18 spin columns and resuspended at 1 µM with 0.1% formic acid for LC–MS analysis.

Chromatographic separations were performed using a Dionex Ultimate 3,000 RPLC nanoLC system configured for preconcentration. Integrafrit trap columns (30 × 0.1 mm) and picofrit analytical columns (20 × 0.075 cm) were packed in-house using 3.5 µM Waters Xbridge BEH C18 (Milford, MA). Peptides were loaded onto the trap column in aqueous solvent containing 2% acetonitrile and 0.1% formic acid for 5 min at a flow rate of 5 µl min$^{-1}$. Water (mobile phase A) and acetonitrile (mobile phase B), each containing 0.1% formic acid, were used with a linear gradient of 2–40% B over 60 min at a flow rate of 300 nl min$^{-1}$. All spectra were acquired in the Orbitrap analyzer of an Orbitrap Fusion Lumos Tribrid mass spectrometer (Thermo Fisher Scientific, San Jose, CA). Higher-energy collisional dissociation (HCD) was performed using a normalized collisional energy (NCD) of 35% in a 3 ms top speed data-dependent method. Dynamic exclusion was enabled with an exclusion duration of 8.00 s. MS1 (from m/z 400–2,000) and MS2 spectra were collected at resolving powers of 60 K and 15 K (at m/z 200), respectively.

Proteome Discoverer 2.0 with Sequest HT and ptmRS site localization software was used to search all data against a forward and reverse FASTA database containing the entire D. melanogaster proteome. All searches included phosphorylation of serine, threonine, and tyrosine as a variable modification and carbamidomethylation of cysteine as a fixed modification. Mass tolerances of 10 p.p.m. and 0.02 Da were used for precursor and fragment ions, respectively. Product ions typically observed during collisional activation (a, b, and y-type) were considered for spectrum matching. A fixed value PSM validator filtered matches based on a maximum delta C$_n$ of 0.05. For both PSMs and peptides, strict and relaxed target FDR settings were 0.01 and 0.05, respectively. PtmRS operating in PhosphoRS mode was used for phosphorylation site localization in which identified sites with isoform confidence probability less than 99% were further inspected manually.

**Native gel electrophoresis.** MBP-CTD fusions were diluted in 2X CN-PAGE buffer (25% glycerol, 62.5 mM Tris-HCl pH 6.8, 1% bromophenol blue) loaded onto 6% clear native-PAGE gels (SDS–PAGE gels with SDS omitted and a acrylamide/bisacrylamide ratio of 29:1) and electrophoresed in running buffer (25 mM Tris-HCl, 192 mM glycine).

**Size exclusion chromatography.** A 100 ul protein standard mix containing thyroglobulin, apoferritin, aldolase, ovalbumin, and RNAseA, with stokes radii of 85, 67.1, 48.1, 30.5 and 16.4 Å, respectively or MBP or MBP-CTD6HIS fusion proteins, were loaded onto a Superose 6 10/300 column at 0.2 ml min$^{-1}$ in CTD buffer at 4 °C. Elution was monitored by UV absorbance and the identity of the protein in each peak was verified by SDS–PAGE. Standards were run before and after MBP-CTD fusions were analysed. The standard curve was generated by plotting $\sqrt{-\log K_{av}}$ versus $R_s$ for the average of three standard runs, where $K_{av} = V_e - V_0/V_c - V_o$ ($V_e$ is elution volume, $V_o$ is void volume as determined by the elution of blue dextran and $V_c$ is the total column volume).

**Small angle X-ray scattering.** SAXS data ($I(q)$ versus $q$, where $q = 4\pi\sin/\lambda$) were collected at the G1 beamline at the Cornell High Energy Synchrotron Source with an X-ray beam energy of 9.944 keV using dual Pilatus 100K-S detectors. Samples were exposed for 20 × 1 s frames. For each sample and at every concentration, buffer blanks were collected immediately before and after sample data acquisition. Data acquisition, buffer subtraction, data reduction, Guinier analysis to a $qR_g < 1.1$, and molecular weight determination by extrapolation to I(0) and comparison to a glucose isomerase control were carried out using BioXtas RAW[29] and plotted in Matlab. Buffer subtracted averaged scattering curves collected at three concentrations were averaged using PRIMUS, and distance distribution functions, $R_g$ and $D_{max}$ were calculated using AUTOGNOM[48]. Theoretical scattering data for MBP was generated using CRYSOL from 1ANF.pdb and plotted using AUTOGNOM. The Flory equation used to predict the $R_g$ of the denatured CTD is, $R_g = R_0 N^v$, where $R_0 = 1.927$, $v = 0.598$ (ref. 49) and $N = 318$, the number of amino acids in the D.melCTD portion of the MBP-D.melCTD fusion, including the C-terminal hexahistidine tag. Ensemble optimization models were generated using EOM 2.1, with averaged scattering curves and 1ANF.pdb as inputs[34,35]. Best agreement of $R_g$ for EOM derived ensembles and the experimentally obtained data were obtained using a compact chain and native-like chain options for the apo and phospho CTD, respectively. Final ensembles were visualized using PyMol, oriented by MBP, and displayed adjacent to the structure of RNA polymerase II from 1Y1W.pdb for comparison[50]. Flexibility analysis was carried out using ScÅtter version 2.3 (ref. 43) and plotted in Excel or RAW.

**Limited proteolysis.** MBP-D.melCTDFLAG-STREP or MBP-H.sapCTD6HIS were end labelled on the terminal CTD repeat with casein kinase II (New England

Biolabs) in protein kinase buffer (50 mM Tris-HCl pH 7.5, 10 mM MgCl₂, 0.1 mM EDTA, 2 mM DTT, 0.01% Brij 35), 0.5 mM ATP and $^{32}$PγATP(MP Biomedical) for 30 min at 30 °C, and then buffer exchanged into CTD buffer using a BioRad BioSpin6 column. To reduce background signal from auto-labelled CKII in the MBP-*H.sap*CTD6HIS experiments, CKII was first pre-incubated in kinase buffer with 4 mM ATP, in the absence of $^{32}$PγATP, for three h at 30 °C. This pre-incubation reaction was then added to a MBP-*H.sap*CTD6HIS labelling reaction resulting in final reaction conditions as described above for labelling MBP-*D.mel*CTDFLAG-STREP. End labelled CTD was digested in CTD buffer for 30 s at room temperature by adding protease concentrations ranging from 2 ng to 6,000 ng per 12 μl reaction. Reactions were quenched by the addition of PMSF to a final concentration of 16 mM and flash frozen on liquid nitrogen. Reactions were thawed in SDS-PAGE sample buffer and resolved in 15% SDS-PAGE gels and exposed to a phosphoimager. For comparison of digestion of apo and phospho CTD, labelling and buffer exchange were carried out as described above, and aliquots of equal volume were phosphorylated by P-TEFb overnight at 24 °C in CTD buffer containing 1 mM ATP and 1/10th reaction volume of Protein Kinase Buffer (NEB) or subject to a mock kinase reaction of identical composition but lacking P-TEFb. The P-TEFb and mock reactions were mixed and immediately subject to limited proteolysis as described above, and fragments were resolved on a 12% SDS-PAGE gel. Signal intensity of the full length proteins were quantified using ImageJ and data from three experimental replicates, representing two separate labelling and kinase reactions, were plotted as the percentage of remaining undigested signal relative to the no protease control at each protein concentration.

**Drosophila CTD sequence alignments.** Amino acid sequences of the Pol II CTD for twelve *Drosophila* species, beginning with the sequence YSPTSPNYTAS were aligned using MUSCLE[51] improved manually, and visualized using BOXSHADE.

**Fly strains and lethality test.** UAS-Rpb1i and yw; Act-GAL4/CyO and C5-Gal4 fly lines were obtained from the Bloomington Stock Center (BDSC 36830, 4414, and 30839, respectively). Sequences encoding the Rpb1$^{WT}$, Rpb1$^{W81}$ or Rpb1$^{hu}$ with double FLAG-tags at the C-termini were subcloned into the pUASt-attB vector, followed by transformation into the *PhiC31 attP 86Fb* strain[52]. Rpb1i-resistance of the ectopically expressed Rpb1 variants was achieved by changing the part of the coding sequence of Rpb1 that is targeted by the 21 nucleotide shRNA (sense strand: 5′-AACGGTGAAACTGTCGAACAA-3′) to 5′-AACCGTCAAGTTGAGCAACAA. The UAS-Rpb1i, UAS-Rpb1 lines were generated by routine matings and meiotic recombination. Rescue tests were done by mating virgin female yw; Act-GAL4/CyO (for ubiquitous expression) or C5-GAL4 (for wing expression) to male UAS-Rpb1i, UAS-Rpb1$^{WT or mut}$. Animals were raised at 24 °C for wing experiments or 21 °C for adult fly rescue experiments. For wing experiments, at least 50 individual offspring were examined and photographs were taken of representative individuals. For adult rescue experiments, rescue was assessed by the emergence among the progeny of straight-wing adults (Act-GAL4/ + ; UAS-Rpb1i, UAS-Rpb1$^{WT or mut}$/ + ) or adults with normal wing growth (C5-GAL4/ UAS-Rpb1i, UAS-Rpb1$^{WT or mut}$).

Extent of endogenous Rpb1 knockdown was assessed by measuring endogenous Rpb1 or ectopically expressed Rpb1 using qPCR. At least three groups of 4–10 rescue animals or offspring from yw control crosses were collected and RNA was isolated using 100 ul of Trizol reagent (Invitrogen). cDNA was prepared using MMLV reverse transcriptase (Promega) with random hexamers and oligodT. cDNA was used to template qPCR reactions with a forward primer hybridizing to the 21 nt region of wild-type or ectopically expressed Rpb1 mutated to generate RNAi resistance, and a reverse primer 5′-GCCTCCAGTTCCTGGATG-3′. Levels of endogenous Rpb1 were normalized to Actin expression levels (primers 5′-TCA GTCGGTTTATTCCAGTCATTCC-3′ and 5′-CCAGAGCAGCAACTTCTTCG TCA-3′) and displayed as percentage of endogenous Rpb1 expression for the yw control crosses. Expression of ectopic Rpb1 was assessed by normalizing expression levels to Rp49 (primers 5′-TACAGGCCCAAGATCGTGAA-3′ and 5′-ACGTTGT GCACCAGGAACTT-3′) and displayed as a percentage of expression from UAS-Rpb1i, UAS-Rpb1$^{WT}$rescue animals.

For the genetic complementation assay of the Rpb1 mutant allele (RpII215$^{G0040}$), the w$^{67c23}$ P{lacW}RpII215$^{G0040}$/FM7c fly line was obtained from the Bloomington Stock Center (BDSC 11547). The da-GAL4, UAS-Rpb1 lines were generated by recombining the da-GAL4 transgene from Bloomington stock 55850 with our UAS-Rpb1 transgene. During the course of constructing the da-GAL4, UAS-Rpb1 lines, we found that flies homozygous for da-Gal4, UAS-Rpb1$^{hu}$ were rare so this pair of transgenes was maintained over a TM3 balancer marked with Sb⁻. We have observed this issue with some other UAS-Rpb1 constructs and suspect that doubling the gene dosage of some mutant UAS-Rpb1 transgenes is semi-lethal. In triplicate, 8 virgin females with the G0040 Rpb1 allele were mated to 8 da-GAL4, UAS-Rpb1 males or yw males as a negative control. Flies were raised at 24 °C and 70% humidity and male adult offspring were identified and counted according to the phenotypes describe in the results. For the cross with UAS-Rpb1$^{hu}$ males, TM3-containing progeny were recognized by the stubble bristles and white eyes and not counted.

**Western blot and immunofluorescence.** Western blot analysis for ectopic expression of Rpb1 was done by dissecting pharate adults from the pupal case and then homogenizing and boiling the tissue in LDS sample buffer (Invitrogen). Equal numbers of male and female pharate adults were selected and pupae of the genotype yw; ActGAL4/ + ; UAS-Rpb1i, UAS-Rpb1$^{wt/mut}$/ + were distinguished from the yw; CyO/ + ; UAS-Rpb1i, UAS-Rpb1$^{wt/mut}$/ + counterpart by the intensity of red pigment in the eye (the UAS-Rpb1 and ActGAL-4 transgenes each have an accompanying mini-white gene marker). For western blotting, tissues equivalent to 0.3 pharate adults were loaded into each lane on a 3–8% Tris-Acetate SDS-PAGE gel (Life Technologies). Transgenic CTD expression was detected with mouse anti-FLAG M2 antibody (1:1,000; Sigma). Spt5 was detected with rabbit anti-Spt5 antibody (1:3,000). The blot was subsequently probed with goat anti-rabbit IgG (1:3,000; Alexa Fluor 488) and goat anti-mouse IgG (1:3,000; Alexa Fluor 647) and visualized with a Typhoon (GE Healthcare).

Salivary glands from third instar larvae from the same cross as described in the RNAi lethality test were dissected and squashed as previously described with[53], and incubated first with anti-FLAG M2 monoclonal antibody (Sigma Aldrich) and anti-Rpb3 antisera overnight at 4 °C, and then with goat anti-mouse Alexa Fluor 568 conjugated IgG and goat anti-rabbit Alexa Fluor 488 conjugated IgG secondary antibodies for 3hr at room temperature. DNA was stained with Hoescht dye. Images from multiple channels were overlaid in Photoshop.

**Data availability.** The data that support the findings of this study are available from the corresponding author on request.

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

## Acknowledgements

We thank Roderico Acevedo for help with SAXS data collection, Richard Gillilan for his help at CHESS and John T. Lis for contributing the P-TEFb virus. We thank Xiangyun Qiu for assistance with SAXS data analysis. This work is based on research conducted at the Cornell High Energy Synchrotron Source (CHESS), which is supported by the National Science Foundation and the National Institutes of Health/National Institute of General Medical Sciences under NSF award DMR-0936384, using the Macromolecular Diffraction at CHESS (MacCHESS) facility, which is supported by award GM-103485 from the National Institutes of Health, through its National Institute of General Medical Sciences. We thank the TRiP at Harvard Medical School (NIH/NIGMS R01-GM084947) for providing transgenic RNAi fly stocks used in this study. This work was supported by GM047477 to D.S.G., NSF grant MCB-1515974 to S.A.S. B.P. was supported in part by NIAID T32 AIO74451. Mass spectrometry work is supported by grants from the National Institutes of Health (R01 GM104896 to Y.J.Z. and R21EB018391 to J.S.B.) and Welch Foundation (F-1778 to Y.J.Z. and F-1155 to J.S.B.). Funding from the University of Texas System for support of the University of Texas System Proteomics Core Facility Network is gratefully acknowledged.

## Author contributions

B.P. designed experiments, analysed data, performed the purifications, SEC, SAXS, limited proteolysis and immunofluorescence experiments, and wrote the manuscript. F.L. generated transgenic flies, designed and conducted RNAi rescue experiments, performed the western blot, and commented on the manuscript. E.B.G. and S.A.S. designed experiments, analysed data, and commented on the manuscript. J.E.M., M.R.M., Y.J.Z. and J.S.B. designed, carried out and analysed mass spectrometry experiments. D.S.G. designed experiments, performed RNAi rescue and complementation experiments, analysed data, and prepared the manuscript.

## Additional information

**Competing interests:** The authors declare no competing financial interests.

**Publisher's note**: 

