## [Peer Review File · Nature Communications]

Reviewers' Comments:

Reviewer #1 (Remarks to the Author):

In this manuscript Gilmour and colleagues report the results of two disparate lines of investigation that make the same point - the sequence of the CTD is less important than its overall structure. The strongest part of the paper describes a genetic swap where the human CTD is used to replace the drosophila as the only (or primary) form of Pol II in the organism. If true, this is a stunning result and makes this paper worthy of publication on its own. The western blot in figure 7d is interpreted to show a lack of expression of the WT Drosophila Rpb1 subunit but the gel is smeary and it is not clear to what extent expression of the endogenous Rpb1 has been reduced. A quantitative rtPCR or Northern blot of the endogenous Rpb1 mRNA would go a long way to support the author's contention that the endogenous Rpb1 is not expressed.

The structural studies are less compelling but do support the general idea that the human and Drosophila CTD are structurally similar. Gel filtration and low angle light scattering support a structural model similar to other intrinsically disordered proteins. The model presented offers an interesting perspective on CTD structure. These results are somewhat limited, however, by the inclusion of the maltose-binding protein. This protein is known to stabilize otherwise unstable protein and this is likely through interacting with the fusion partner. This might influence the results in several ways. First, the Stokes radius of the fusion is smaller than predicted. Could this be due to interactions between the CTD and MBP?

The most confusing part of this analysis is the protease sensitivity data. It is not clear to me why there should be a protease sensitive site about 10 kD from the C-terminus. Might this be due to interactions with MBP? The authors should do a protease sensitivity assay on the excised CTD to determine whether the hypersensitive site is an intrinsic property of the CTD.

Finally, there are several minor suggestions.

1. In Figure 1d where does MBP run?

2. Figure 4g needs size markers.

3. How many residues are phosphorylated by P-TEFb in Figures 1 and 2? Are these Ser2, Ser5 or both?

4. Why doesn't SDS make cleavage random like predicted in Figure 4a. Is there some residual structure in SDS or does the particular sequence of the CTD effect recognition of protease cleavage sites?

If these concerns are adequately addressed the paper would be strengthened and would represent a very significant contribution to our understanding of CTD structure and function.

Reviewer #2 (Remarks to the Author):

The article by Portz, et al., reports the characterization of the RNA polymerase II C-terminal domain (CTD) from drosophila and humans using sequence analysis, size exclusion chromatography, SAXS, proteolytic accessibility and, finally, in vivo CTD swap experiments in drosophila. The data in the paper appear to be of very high quality and true. Several novel insights have emerged from the authors' studies. First, the CTDs from both drosophila and humans adopt somewhat compact, disordered conformations as documented by size exclusion and SAXS data. Second, phosphorylation causes slight expansion of one of the CTD ensembles relative to the unmodified form. Third, proteolytic mapping showed that, despite the presentation of numerous protease sites within the repetitive heptad motifs, relatively few sites were actually cleaved by proteases, suggesting conformational heterogeneity of these heptads within the CTDs. The conformational heterogeneity was apparent in protease maps for both the drosophila and human CTDs despite sequence differences. Phosphorylation enhanced proteolytic cleavage, possibly due to increased accessibility of cleavage sites as suggested by the SAXS results. Finally, using genetically manipulated flies, the authors demonstrated that the human CTD functionally substituted for the fly homolog. Together, the results strongly suggest that the drosophila and human CTDs, despite some extent of sequence differences, have conserved structural features that support conserved functions in the two organisms. While being comprised of dozens of repeats of closely related heptad motifs, the proteolysis results strongly suggest that the local conformational features of the two different CTDs are similarly heterogeneous. These results provide novel insights into the disordered conformational features of the RNA Pol II CTD and consequently the manuscript warrants consideration at Nature Communications. Some issues

concerning the phosphorylation and ensemble generation experiments need to be addressed before a final decision could be reached. .

1. What is the stoichiometry of CTD phosphorylation by P-TEFb in the authors' in vitro experiments? The authors should relate their results on CTD phosphorylation to the recent results from Suh, et al., and Schüller, et al., in Mol Cell (January, 2016) showing that Pol II CTD is incompletely phosphorylated in vivo. Is the level of phosphorylation in the current in vitro study reflective of the in vivo results? What is the stoichiometry of phosphorylation in the current study? Have the authors considered that the heterogeneous exposure of protease sites in vitro may be relevant to the observation of partial phosphorylation in vivo?

2. Regarding the ensemble calculations, what are the conformational properties of the CTD molecules, both apo and phosphorylated, within the ensembles? It is routine in the IDP field now to use both NMR and SAXS data to define the ensemble-average local and global conformational features, respectively, of molecules within ensembles, but here only SAXS data is being used. This means that the local conformational features have been completely determined by the ensemble generation algorithm. How are proline isomers handled in these computations? Are the cis/trans isomer ratios similar or different for the apo and phosphorylated CTD ensembles? Are the similarities or differences justified based on any experimental data other than SAXS? How is phosphorylation handled in the algorithm? What types of phosphate interactions are considered? Aside from producing ensembles that recapitulate the experimental Rg values from SAXS, what arguments can the authors make that the ensembles are true? What are the populations of beta-strand versus alpha-helix psi/phi values for the different ensembles? Are the results consistent with the authors' statements in the introduction regarding low populations of beta-turn conformers?

Reviewer #3 (Remarks to the Author):

Portz et al. present a novel analysis of the structure of C-terminal domain (CTD) of the RNA polymerase II large subunit. It has long been realized that the CTD is a highly important yet relatively disordered binding site for regulators of polymerase function and that its activity is modified by phosphorylation events. There are a number of models for the structure of the CTD based on prior work. The authors show compelling evidence for a new model. They show that the structure is not fully disordered as some models have suggested. Instead it has a measurable compaction, as shown by gel filtration and SAXS. They also show that phosphorylation

decompacts the structure somewhat. There is a dramatic localized sensitivity to proteolytic cleavage that the authors interpret as evidence for heterogeneity in the structure. I can think of no alternative explanation and find the result intriguing. This heterogeneity is seen in fly and human CTD, suggesting that it is a conserved feature. Finally, the authors present evidence that human CTD can functionally replace the fly CTD in transgenic *Drosophila*. Since there has been much speculation that CTD sequence differences between phyla could be due to different binding of regulators, the apparent function of the human sequence in the fly is eye opening.

The work is a highly significant contribution to our understanding of a critical and much studied component of transcriptional control in animals. The paper is disarmingly simple, yet will alter perceptions of CTD greatly. For the most part it is well and clearly written. I do have one significant concern, though, as well as a minor suggestion.

1. The section showing that human CDT functions in the fly is too briefly written and challenging to follow and should be rewritten to clarify. Add explanatory sentences.

The description of the rescue of depletion in the wing is clear, but it would be more rigorous to give some quantification of the results. How many flies were examined? Was there any variation in the phenotype. Only single representative pictures are given.

As I understand it, the RNAi knockdown of the large subunit of endogenous RNA polymerase is only partial. Yet the authors seem to be suggesting that there is a complete replacement in the lethality rescue experiments that use ACTIN GAL4. Fig. 7d is said to show that there is no detectable endogenous fly large subunit when the human CDT variant is present (along with the RNAi transgene). I do not find this western compelling. There is a smear of signal below the human CDT variant that is as strong as the signal for the endogenous fly protein in the yw control. If I am miss interpreting this result, please describe more carefully. Or use a different antibody as one really ought to be have better signal to noise with an RNA polymerase western.

I do not find the polyteen staining in Fig. 7 e compelling either. It is claimed that the human CTD variant co localizes with endogenous RNA polymerase. The resolution is low, but it is clear that there are many green and red stained areas of the chromosome as well as yellow, co staining areas. It is fair to say that the same is true for the FLAG tagged fly CTD variant. I wonder if the FLAG tagged fly and human CDT variants co localize with each other but not exactly with

Rpb3. Perhaps this could be seen with high resolution images comparing Rpb3/FLAG human and Rpb3/FLAG fly. If these two patterns were the same, that would be a strong argument.

I wonder if it may be more prudent to conclude, not that the results show that the human CTD encodes all functions needed to rescue the fly enzyme, but that it encodes sufficient that it can at least partially rescue many function and perhaps may fully rescue all.

2. My minor comment is that the introduction is on the long side. I appreciated its scholarly description of prior work, but it reads like a detailed review. The same set of papers and their essential conclusions could be captured well in about 70% of the length.

Responses to reviewer comments are in bold.

Reviewer #1 (Remarks to the Author):

In this manuscript Gilmour and colleagues report the results of two disparate lines of investigation that make the same point the sequence of the CTD is less important than its overall structure. The strongest part of the paper describes a genetic swap where the human CTD is used to replace the drosophila as the only (or primary) form of Pol II in the organism. If true, this is a stunning result and makes this paper worthy of publication on its own. The western blot in figure 7d is interpreted to show a lack of expression of the WT Drosophila Rpb1 subunit but the gel is smeary and it is not clear to what extent expression of the endogenous Rpb1 has been reduced. A quantitative rtPCR or Northern blot of the endogenous Rpb1 mRNA would go a long way to support the author's contention that the endogenous Rpb1 is not expressed.

Thank you for your support of the genetics experiments. We agree the result warrants publication and understand the concern, originating with the quality of the western blot, that our knockdown of endogenous Rpb1 may be incomplete. We have altered the writing to acknowledge this and removed the western for endogenous Pol II. Given that RNAi knockdowns are not complete, we now provide an independent genetic complementation assay. In Fig. 7e, we show that ectopic expression of Rpb1 harboring either the normal Drosophila CTD or the human CTD complements an early embryonic lethal Rpb1 allele to allow development to adulthood.

The structural studies are less compelling but do support the general idea that the human and Drosophila CTD are structurally similar. Gel filtration and low angle light scattering support a structural model similar to other intrinsically disordered proteins. The model presented offers an interesting perspective on CTD structure. These results are somewhat limited, however, by the inclusion of the maltose binding protein. This protein is known to stabilize otherwise unstable protein and this is likely through interacting with the fusion partner. This might influence the results in several ways. First, the Stokes radius of the fusion is smaller than predicted. Could this be due to interactions between the CTD and MBP?

The concern that MBP may be interacting with the CTD is valid and we added experiments to rule out this possibility, though we disagree with the notion that MBP stabilizes proteins via interaction with the fusion partner. MBP is used to stabilize proteins and aid in structural determination using X-ray crystallography, and there are now over 100 structures of proteins fused to MBP in the Protein Data Bank, 24 of which are available in both the fused and un-fused form, with no appreciable differences (for review, see David S. Waugh. Protein Science 25:559-571 2016). Further, the CTD has evolved to extend from a globular and acidic Pol II enzyme, and we wanted to study the CTD in a context mimicking its natural position C—terminal to the globular and acidic MBP in a way that was tractable for our studies. Nevertheless, we recognize the possibility that MBP:CTD interactions could alter the interpretation of the data. To ensure this wasn't the case, we added supplementary figure 4d. In this experiment, we fuse the CTD to GST instead of MBP, and repeat our limited proteolysis assay. The result of this experiment is identical to that obtained from the MBP-CTD, strongly suggesting CTD:fusion protein interaction is not occurring. Additionally, the Pr vs r plot of the MBP-CTD fusion from our SAXS studies is indicative of a rod-like extension emanating from a globular domain, which argues against significant MBP:CTD interaction.

The most confusing part of this analysis is the protease sensitivity data. It is not clear to me why there should be a protease sensitive site about 10 kD from the C-terminus. Might this be due to interactions with MBP?

As mentioned above, we address this valid concern in Supplementary Fig. 4d. Also, the array of different proteases, with a range of specificities, all giving rise to similar cleavage patterns, strongly suggests the distal hypersensitive site is an intrinsic feature of the CTD.

The authors should do a protease sensitivity assay on the excised CTD to determine whether the hypersensitive site is an intrinsic property of the CTD.

Since the CTD has evolved to extend from a globular and acidic protein, we believe the globular and acidic MBP provides an experimentally tractable proxy for the globular and acidic body of Pol II. We also wanted to ensure we were studying soluble and monomeric CTD, which is aided by MBP.

Our new data generated on a GST-CTD fusion argues a discontinuous pattern of protease sensitivity

is an intrinsic property of the CTD and not an artifact of interaction with the fusion protein.

Finally, there are several minor suggestions.

1. In Figure 1d where does MBP run?

This has been added in Supplementary Fig. 1b, in which we compare directly the MBP and MBP-CTD fusions with the protein standards. The globular protein standard curve, and the CTD fusion curve, intersect near MBP. The legend has been changed to reflect this data.

2. Figure 4g needs size markers.

We repeated the human CTD limited proteolysis experiment and include the CTD size markers, as well as a direct comparison to the cleavage pattern of the *Drosophila* CTD. Additionally, we altered our MBP-*H.sap*CTD labeling conditions to greatly improve the signal to noise in this experiment. We also used a MBP-*H.sap*CTD fusion that lacked a TEV protease site for the limited proteolysis, so we are now confident that the MBP-proximal hypersensitive site is a feature of the CTD, not cleavage of the TEV recognition sequence by proteinase K. The results and methods sections have been changed to reflect these changes

3. How many residues are phosphorylated by PTEFb in Figures 1 and 2? Are these Ser2, Ser5 or both?

To address these interesting questions, we added mass spectrometry data to determine the amount of phosphorylation, as well as the location of many of the phosphates. We have added MALDI-TOF mass spectrometry data in Supplementary Fig 2 a and b that suggests approximately 25 phosphates are added to the CTD, in excellent agreement with estimates from our SAXS data. We have also added HCD mass spectrometry data in Supplementary Fig. 2c that shows a preference for incorporation of phosphates at the Ser5 position, which is in agreement with literature we now cite in the results section. We detect Ser2 phosphorylation only in those heptads lacking an SP motif in the 5-6 position.

4. Why doesn't SDS make cleavage random like predicted in Figure 4a.

Is there some residual structure in SDS or does the particular sequence of the CTD effect recognition of protease cleavage sites?

At the concentrations of SDS compatible with protease activity, there is likely to be residual structure. This is evident by the lack of protease sensitivity in 0.1% SDS in the maltose binding protein itself, which would manifest as radiolabeled bands running above the CTD ladder. At 0.5% SDS, one band above the CTD ladder becomes more evident, suggesting the conditions are sufficiently denaturing to alter the conformation of MBP, but not completely denaturing as to render the entire MBP or CTD uniformly sensitive to proteolysis.

If these concerns are adequately addressed the paper would be strengthened and would represent a very significant contribution to our understanding of CTD structure and function.

We agree with your evaluation that the work represents a very significant contribution to our understanding of CTD structure and function, We have addressed every concern with new experimental data as well as attempts to clarify the presentation. Thank you for your fair and supportive assessment of the work.

Email Reviewer #2 (Remarks to the Author):

The article by Portz, et al., reports the characterization of the RNA polymerase II C-terminal domain (CTD) from *Drosophila* and humans using sequence analysis, size exclusion chromatography, SAXS, proteolytic accessibility and, finally, in vivo CTD swap experiments in *Drosophila*. The data in the paper appear to be of very high quality and true. Several novel insights have emerged from the authors' studies. First, the CTDs from both *Drosophila* and humans adopt somewhat compact, disordered conformations as documented by size exclusion and SAXS data. Second, phosphorylation causes slight expansion of one of the CTD ensembles relative to the unmodified form. Third, proteolytic mapping showed that, despite the presentation of numerous protease sites within the repetitive heptad motifs, relatively few sites were actually cleaved by proteases, suggesting conformational heterogeneity of these heptads within the CTDs. The conformational heterogeneity was apparent in protease maps for both the *Drosophila* and human CTDs despite sequence differences. Phosphorylation enhanced proteolytic cleavage, possibly due to increased accessibility of cleavage sites as suggested by the SAXS results. Finally, using genetically manipulated flies, the authors demonstrated that the human CTD functionally substituted for the fly homolog. Together, the results strongly suggest that the *Drosophila* and human CTDs, despite some extent of sequence differences, have conserved structural features that support conserved functions in the two organisms.

While being comprised of dozens of repeats of closely related heptad motifs, the proteolysis results strongly suggest that the local conformational features of the two different CTDs are similarly heterogeneous. These results provide novel insights into the disordered conformational features of the RNA Pol II CTD and consequently the manuscript warrants consideration at Nature Communications. Some issues concerning the phosphorylation and ensemble generation experiments need to be addressed before a final decision could be reached. .

Thank you for your support of the work. We agree that the structural studies and our discovery of structural heterogeneity is novel. We thank you for your insightful questions and address them below.

1. What is the stoichiometry of CTD phosphorylation by PTEFb in the authors' in vitro experiments? The authors should relate their results on CTD phosphorylation to the recent results from Suh, et al., and Schüller, et al., in Mol Cell (January, 2016) showing that Pol II CTD is incompletely phosphorylated in vivo. Is the level of phosphorylation in the current in vitro study reflective of the in vivo results? What is the stoichiometry of phosphorylation in the current study? Have the authors considered that the heterogeneous exposure of protease sites in vitro may be relevant to the observation of partial phosphorylation in vivo?

We have added MALDI-TOF mass spectrometry data to the paper in Supplementary Fig. 2 a and b. This data reveals a change in mass of 1.9kD as a function of phosphorylation, which is in excellent agreement with the change in mass we measured by the Guinier fit of the SAXS data. Together, these data support the addition of about ~25 phosphates to the CTD in our in vitro kinase conditions, or less than 1 phosphate per heptad. This is in close agreement with observations in both Suh et al. and Schuller et al. papers, which show incomplete CTD phosphorylation. We have also added HCD mass spectrometry data in Supplementary Fig. 2c identifying the location of many of the incorporated phosphates, and observe that heptads are phosphorylated only once, which also agrees with Suh et al. and Schuller et al. We have added this information to the results and cited the two papers as per your suggestion.

We have consider the possibility that structural heterogeneity in the CTDs of metazoans could localize both the binding of factors, and the application of post translational modifications to particular regions of the CTD as mentioned in the discussion section. Currently, our limited proteolysis experiments are done on a short timescale (30s), whereas our kinase assays to generate phospho-CTD for subsequent analysis are overnight reactions or longer. Thus, we hesitate to directly relate structural heterogeneity with the localization of phosphates in our assays. However, we agree that structural heterogeneity across the CTD could in theory help regulate the localization of binding partners or post-translational modifications.

2. Regarding the ensemble calculations, what are the conformational properties of the CTD molecules, both apo and phosphorylated, within the ensembles? It is routine in the IDP field now to use both NMR and SAXS data to define the ensemble average local and global conformational features, respectively, of molecules within ensembles, but here only SAXS data is being used. This means that the local conformational features have been completely determined by the ensemble generation algorithm. How are proline isomers handled in these computations? Are the cis/trans isomer ratios similar or different for the apo and phosphorylated CTD ensembles? Are the similarities or differences justified based on any experimental data other than SAXS? How is phosphorylation handled in the algorithm? What types of phosphate interactions are considered? Aside from producing ensembles that recapitulate the experimental Rg values from SAXS, what arguments can the authors make that the ensembles are true? What are the populations of beta strand versus alpha helix psi/phi values for the different ensembles? Are the results consistent with the authors' statements in the introduction regarding low populations of beta turn conformers?

It is true that NMR data would provide additional local structural constraints to our models that would improve their ability to describe the CTD ensemble. However, we feel that NMR on the full-length CTD would be a technical breakthrough in it's own right and outside the scope of our current studies, which already include biophysical, biochemical, and genetic experiments. However, the nature of the *Drosophila* CTD sequence means it could be a tractable system for NMR, unlike the human and yeast CTD which contain long stretches of identically repeating sequence. NMR characterization of the CTD remains an ongoing goal of the research program. We feel that this work

establishes the *Drosophila* CTD as a tractable experimental model system for biophysics, biochemistry and genetic, to include NMR in the future.

You comments have motivated us to alter the description of the models to aid in their interpretation. We now more clearly articulate the limitations of the models in the paper. With our models, we sought to provide readers, particularly the gene regulation/transcription audience, a conceptualization of the size of the CTD relative to the catalytic core of Pol II. Various models for the CTD structure, addressed in our introduction, have been proposed and the CTD has been rendered in a variety of ways in the literature. Our models have the advantage that they are constrained by Rg measurements obtained on monomeric, full-length CTD under physiologically relevant buffer conditions. They lack sufficient data to make interpretations about changes in local structure, proline isomerization, or phi/psi angles that could be induced by phosphorylation.

Reviewer #3 (Remarks to the Author):

Portz et al. present a novel analysis of the structure of C terminal domain (CTD) of the RNA polymerase II large subunit. It has long been realized that the CTD is a highly important yet relatively disordered binding site for regulators of polymerase function and that its activity is modified by phosphorylation events. There are a number of models for the structure of the CTD based on prior work. The authors show compelling evidence for a new model. They show that the structure is not fully disordered as some models have suggested. Instead it has a measurable compaction, as shown by gel filtration and SAXS. They also show that phosphorylation decompacts the structure somewhat. There is a dramatic localized sensitivity to proteolytic cleavage that the authors interpret as evidence for heterogeneity in the structure. I can think of no alternative explanation and find the result intriguing. This heterogeneity is seen in fly and human CTD, suggesting that it is a conserved feature. Finally, the authors present evidence that human CTD can functionally replace the fly CTD in transgenic *Drosophila*. Since there has been much speculation that CTD sequence differences between phyla could be due to different binding of regulators, the apparent function of the human sequence in the fly is eye opening.

The work is a highly significant contribution to our understanding of a critical and much studied component of transcriptional control in animals. The paper is disarmingly simple, yet will alter perceptions of CTD greatly. For the most part it is well and clearly written. I do have one significant concern, though, as well as a minor suggestion.

We sought to distill the findings as simply as possible and appreciate your description of the paper as “disarmingly simple, yet will alter perceptions of the CTD greatly.” We feel the work, which employs biophysical, biochemical, and genetic approaches will appeal to readers studying transcription as well as the structure, function and evolution of disordered proteins and wanted each section to be accessible to all audiences.

1. The section showing that human CDT functions in the fly is too briefly written and challenging to follow and should be rewritten to clarify. Add explanatory sentences. The description of the rescue of depletion in the wing is clear, but it would be more rigorous to give some quantification of the results. How many flies were examined? Was there any variation in the phenotype. Only single representative pictures are given.

Based on your suggestion, we have re-written the section describing the rescue of RNAi depletion to add explanatory sentences and more clearly explain the results. We have also further validated the results with an RNAi independent, genetic complementation assay found in Fig. 7e.

For the wing experiments, at least 50 individuals from each cross were examined, and representative photos were taken. There was little phenotypic variation among offspring from the individual crosses. This information has been added to the legend of Supplementary Fig. 6 and the related methods section.

As I understand it, the RNAi knockdown of the large subunit of endogenous RNA polymerase is only partial. Yet the authors seem to be suggesting that there is a complete replacement in the lethality rescue experiments that use ACTIN GAL4. Fig. 7d is said to show that there is no detectable endogenous fly large subunit when the human CDT variant is present (along with the RNAi transgene). I do not find this western compelling. There is a smear of signal below

the human CDT variant that is as strong as the signal for the endogenous fly protein in the yw control. If I am misinterpreting this result, please describe more carefully. Or use a different antibody as one really ought to be able to have better signal to noise with an RNA polymerase western.

We address the concern of incomplete knockdown in a number of ways. First, we altered the writing so as not to suggest a complete replacement of endogenous Rpb1 with an ectopically expressed variant, and have removed the western blot from 7d that was unclear. The quality of the western blot from pupae and adult flies is compromised by highly abundant co-migrating proteins, and using tissues from earlier developmental stages that provide cleaner western blots is not possible as it is difficult to determine the genotype of the offspring at earlier stages of development. To address the concern of incomplete knockdown, we added an additional genetic experiment that does not rely on RNAi. In this experiment we complement an Rpb1 mutant allele that is lethal in early embryogenesis. We show in Fig. 7e that ectopically expressed Rpb1 harboring the endogenous CTD or the human CTD both complement the early embryonic lethal Rpb1 allele and support the development of adult flies. Our ability to complement an early embryonic lethal mutant with the human CTD is an additional and stringent validation of the rescue results obtained using RNAi depletion in both the wing and in entire flies.

I do not find the polytene staining in Fig. 7e compelling either. It is claimed that the human CTD variant co-localizes with endogenous RNA polymerase. The resolution is low, but it is clear that there are many green and red stained areas of the chromosome as well as yellow, co-staining areas. It is fair to say that the same is true for the FLAG tagged fly CTD variant. I wonder if the FLAG tagged fly and human CTD variants co-localize with each other but not exactly with Rpb3. Perhaps this could be seen with high resolution images comparing Rpb3/FLAG human and Rpb3/FLAG fly. If these two patterns were the same, that would be a strong argument.

We have included new high magnification images, obtained from new experiments that show much more co-localization of ectopically expressed Rpb1 and endogenous Rpb3. The crosses and immunofluorescence were carried out as before, but the incubation time with the primary antibodies was extended, and the methods section now reflects this change. Because not all regions of the chromosome are uniformly accessible to antibodies, and not all antibodies are equivalent in affinity, overnight incubation with the primary antibodies provides more uniform staining across chromosomes.

I wonder if it may be more prudent to conclude, not that the results show that the human CTD encodes all functions needed to rescue the fly enzyme, but that it encodes sufficient that it can at least partially rescue many functions and perhaps may fully rescue all.

The writing has been clarified according to your suggestions. We now emphasize that ectopic expression of human Rpb1 can rescue *lethal levels* of RNAi depletion of endogenous Rpb1, and have removed any suggestion of complete replacement.

Also, we incorporate an additional genetic test, independent of RNAi, that supports the conclusion that the human CTD can function in place of the fly CTD to complement an early-embryonically lethal Rpb1 mutant as discussed above.

2. My minor comment is that the introduction is on the long side. I appreciated its scholarly description of prior work, but it reads like a detailed review. The same set of papers and their essential conclusions could be captured well in about 70% of the length.

Each experiment described in the introduction revealed important insights about CTD *function*, but each had caveats with respect to contributing to our understanding of CTD *structure*. We sought to respectfully address those caveats while acknowledging the valuable contributions of the prior work. Our data supports a very different understanding of CTD structure that we felt is best considered in light of the prior literature.

Reviewers' Comments:

Reviewer #1 (Remarks to the Author):

The authors have substantially addressed most of the concerns with the previous submission, especially with regard to the physical studies of CTD structure. I do have lingering concerns over the genetic studies. The authors have included a new experiment designed to assess the function of *Drosophila* Rpb1 harboring a human CTD. This chimeric allele complements a strain in which the WT Rpb1 gene is knocked-down by RNAi. As there may be some residual expression in this strain it was not clear whether the humanized Rpb1 could functionally replace the WT allele. In the new experiment the authors show that Rpb1hu can complement an embryonic lethal Rpb1 mutant (Rpb1G0040). This is an encouraging result but the authors do not give us enough information about the Rpb1G0040 allele to determine whether it is truly an independent test of Rpbhu functionality. As far as I can determine the Rpb1G0040 allele contains a transposon insertion upstream of the Rpb1 coding region. Does this insertion completely knock down Rpb1 expression or is there some low level of expression of WT Rpb1 in this strain. If there is some WT expressed then this new experiment is very similar to the RNAi experiment. The authors should include more information about the embryonic lethal allele. In addition, the authors should emphasize that the Rpb1hu allele can complement but may not be able to completely functionally replace the WT allele.

Reviewer #2 (Remarks to the Author):

The revised manuscript by Portz, et al., is significantly improved over the original submission through clarification of several key issues and through strengthening of the in vivo studies of chimeric CTDs. Issues related to CTD phosphorylation have been addressed using mass spectrometry; new experiments show that the extent of phosphorylation observed in the current studies is similar to that observed in other two recent studies, allowing the new biophysical and biochemical results to be related to these past findings. This enhances the impact of the current studies. Further, the authors now appropriately qualify the structural modeling results that are based on SAXS data. Importantly, they clarify that only the global features of the resulting ensembles are interpretable, and not their residue-level conformational details. Overall, the results in the revised manuscript rigorously establish that the fly and human CTDs have somewhat compact, disordered conformations. Surprisingly, despite being comprised of many, many closely related heptad repeats, and differing in their exact sequences, the fly and human CTDs each exhibit similar conformational heterogeneity that may play roles in function in cells. The observation that the human CTD, despite the noted sequence differences, can substitute for the fly CTD in vivo strengthens the view that the conserved structural features these authors have

identified between the fly and human CTDs are of functional significance. The study also defines the effects of multi-site phosphorylation on CTD structure, showing that the CTD expands slightly through conformational stiffening. Overall, for the first time with full-length CTD samples, this study defines the conformational landscape of the fly and human CTDs as being somewhat compact but intrinsically disordered with local conformational heterogeneity and variable residue accessibility that is modulated by phosphorylation. The study will provide a basis in the future for generating and testing hypotheses regarding the role of these CTD structural features in regulation of transcription by Pol II and its myriad accessory factors. The manuscript is suitable for publication in Nature Communications without further revision.

Reviewer #3 (Remarks to the Author):

The authors have only partially satisfied my concerns, I am afraid. The rewritten section on the analysis of mutant flies is now very clear. The addition of the complementation of an early embryo lethal in polymerase is a first rate idea. However, this potentially compelling experiment is far from complete as presented. The result would be more credible with a larger sample size and an additional control. One is left to judge the significance of seeing 0 flies for the control (wild type parental males not expressing ectopic polymerase) versus 3 or 5 flies in the test cases (parental males expressing polymerase with either a human or fly CTD). I am not an expert on the particular pol II early embryo lethal used, but many mutants in essential genes can be leaky due to perduring maternal product. The rescued flies are reported to be so sickly that they must be helped out of the pupal case. Can we be certain that there were no flies in the controls that might have almost hatched as well? Given this legitimate concern, a much larger sample is needed (N = 100 off spring from each cross). The number of parents crossed should be indicated. It should be clearly stated that no unhatched pupae were apparent from the control cross. In addition, it would be more rigorous to include as a control ectopic expression of an RNA polymerase lacking the CTD. Presumably this mutant would be non functional. The direct side by side is essential to show the rescue is due to the gene in question, not some more general background effect of strain differences.

To be clear, I admire this work and continue to believe that it will significantly impact our understanding of this important domain of RNA polymerase. To have the desired impact, though, it is important that the evidence presented leave no serious room for doubt.

We thank the reviewers for evaluating of our manuscript and feel that our effort to address reviewer's concerns have strengthened our study. Responses to reviewer comments are shown below in italics. Substantial changes to the text are highlighted in yellow in the manuscript.

Reviewer #1 (Remarks to the Author):

The authors have substantially addressed most of the concerns with the previous submission, especially with regard to the physical studies of CTD structure.

We have now addressed your concerns with the genetic studies and our response is outlined below.

I do have lingering concerns over the genetic studies. The authors have included a new experiment designed to assess the function of *Drosophila* Rpb1 harboring a human CTD. This chimeric allele complements a strain in which the WT Rpb1 gene is knocked-down by RNAi. As there may be some residual expression in this strain it was not clear whether the humanized Rpb1 could functionally replace the WT allele.

We now show using qPCR from cDNA made from adult offspring from the RNAi rescue experiments that we knockdown the endogenous Rpb1 transcript to ~5% of the levels found in the yw control (Supplementary Fig 7a). Using qPCR we also show, in agreement with the western blot included, that the humanized Rpb1 is expressed at levels comparable to the ectopically expressed wild-type Rpb1 (Supplementary Fig 7b and c). Together, these new data strengthen the observation that Rpb1 with the human CTD can function in flies to rescue robust and lethal RNAi knockdown of endogenous Rpb1. We have also strengthened the complementation experiment by adding replicates and using new growth conditions, which we outline below.

In the new experiment the authors show that Rpb1hu can complement an embryonic lethal Rpb1 mutant (Rpb1G0040). This is an encouraging result but the authors do not give us enough information about the Rpb1G0040 allele to determine whether it is truly an independent test of Rpb1hu functionality. As far as I can determine the Rpb1G0040 allele contains a transposon insertion upstream of the Rpb1 coding region. Does this insertion completely knock down Rpb1 expression or is there some low level of expression of WT Rpb1 in this strain. If there is some WT expressed then this new experiment is very similar to the RNAi experiment.

Your concern is that minimal residual expression from the p-element containing allele may make this experiment similar to the RNAi rescue experiment. We now show that there is very little endogenous Rpb1 mRNA in our RNAi experiments, ~5% of control levels (Supplementary Fig 7a). Clearly, the humanized Rpb1 provides function, as this level of RNAi knockdown is lethal in the absence of ectopically expressed Rpb1.

We also repeated the complementation assay with each genotype in triplicate. A crucial difference was that we now raise the flies at 24°C with 70% humidity. This change from our prior conditions resulted in more complementing offspring harboring the human CTD, which were also more healthy than the offspring from the prior experiment. Notably, no "escaper" offspring from the yw control cross were observed.

The authors should include more information about the embryonic lethal allele. In addition, the authors should emphasize that the Rpb1hu allele can complement but may not be able to completely functionally replace the WT allele.

We added information explaining the origin of the embryonic lethal line. We also clarified the writing to state definitively our assay is for complementation.

Reviewer #2

Reviewer 2 stated the revised manuscript was “significantly improved” over the initial submission and “is suitable for publication in Nature Communications without further revision.” Thus, we have no further comments and thank reviewer 2 for the helpful suggestions.

Reviewer #3 (Remarks to the Author):

The authors have only partially satisfied my concerns, I am afraid. The rewritten section on the analysis of mutant flies is now very clear. The addition of the complementation of an early embryo lethal in polymerase is a first rate idea. However, this potentially compelling experiment is far from complete as presented. The result would be more credible with a larger sample size and an additional control. One is left to judge the significance of seeing 0 flies for the control (wild type parental males not expressing ectopic polymerase) versus 3 or 5 flies in the test cases (parental males expressing polymerase with either a human or fly CTD). I am not an expert on the particular pol II early embryo lethal used, but many mutants in essential genes can be leaky due to perduring maternal product. The rescued flies are reported to be so sickly that they must be helped out of the pupal case. Can we be certain that there were no flies in the controls that might have almost hatched as well?

We repeated the complementation assay with each genotype in triplicate. A crucial difference was that we now raise the flies at 24°C with 70% humidity. This change from our prior conditions resulted in more complementing offspring harboring the human CTD, which were also more healthy than the offspring from the prior experiment. At the new growth conditions, the complementing flies no longer needed to be pulled from the pupal case, instead they hatched normally. Notably, no “escaper” offspring from the yw control cross were observed from any of the 3 replicates. We have removed the original data from the paper and included only the data from the new crosses, performed in triplicate at 24°C with 70% humidity.

We note that the complementation assay was included to corroborate the RNAi result. In addition to strengthening the complementation assay using triplicate matings under conditions that gave rise to more and healthier complementing offspring, we also strengthened the RNAi experiment by quantifying the degree of knockdown on endogenous Pol II mRNA in adult flies emerging from the RNAi rescue experiments. We observe knockdown to ~5% of normal flies.

Given this legitimate concern, a much larger sample is needed (N = 100 off spring from each cross). The number of parents crossed should be indicated. It should be clearly stated that no unhatched pupae were apparent from the control cross.

Rather than collecting a sample size of 100, we have set the crosses up in triplicate and each time observed complementation by the huRpb1 and no complementation in the absence of ectopically expressed Rpb1; the total numbers of flies from all three crosses are presented in Figure 7e and the number of flies from individual vials are presented in supplemental table 1. In the methods, we include details about the precise growth conditions as well as the number of parents in the crosses. We also highlight in the writing that no offspring hatched from the control cross.

In addition, it would be more rigorous to include as a control ectopic expression of an RNA polymerase lacking the CTD. Presumably this mutant would be non functional. The direct side by side is essential to show the rescue is due to the gene in question, not some more general background effect of strain differences.

This is a good suggestion, but the experiment is impossible because the CTD-less Pol II mutant confers a dominant negative phenotype. Expression of the CTD-less Pol II driven by da-GAL4 is lethal even in the presence of endogenous Pol II. In lieu of this experiment, we feel that the RNAi rescue experiment and the G0040 complementation assay are sufficiently different yet yield corroborating results to rule out the likelihood that the results are due to some strain differences.

The RNAi rescue experiment uses an Actin Gal4 driver, while the G0040 complementation assay uses a daGal4 driver. Moreover, constructing the stocks for each of these experiments involved different mating schemes, which further diminishes the likelihood that strain differences are impacting our results.

To be clear, I admire this work and continue to believe that it will significantly impact our understanding of this important domain of RNA polymerase. To have the desired impact, though, it is important that the evidence presented leave no serious room for doubt.

Thanks for your suggestions; we feel they lead to improved genetics and a more robust conclusion.

Reviewer Comments:

Reviewer #1 (Remarks to the Author):

In this revision the authors have greatly strengthened the genetic complementation assay showing that a transgene expressing a chimeric largest subunit harboring the human CTD can complement a lethal transposon insertion in the endogenous gene. I think that together with the biophysical studies this makes a strong case that some overall structural property rather than a particular CTD sequence supplies the essential CTD function. This paper will make a great impact on future studies of the CTD.

Reviewer #3 (Remarks to the Author):

The revised manuscript thoroughly and completely addresses my concerns. The results presented definitely establish that the human CTD can function effectively in flies. The increased number of flies in the lethality rescue experiments and the improved viability of the rescued flies provide compelling evidence. The demonstration that RNAi reduces wild type RNA polymerase to only 5% of wild type levels clarifies that the human CTD variant of Drosophila RNA pol II must be taking on virtually all of the wild type enzymes functions. This coupled with detailed, innovative structural observations make this a dramatic and important set of observations. The paper has general implications for so called unstructured protein domains as well as for this particular critical component of the basal transcription apparatus. I congratulate the authors on an outstanding piece of work.